# Matrix-Driven Detection and Reconstruction of LLM Weight Homology

## Abstract

Recently, concerns about intellectual property in large language models (LLMs) have grown significantly, particularly around the unattributed reuse or replication of model weights. However, existing methods for detecting LLM weight homology fall short in key areas, including recovering the correspondence between weights and computing significance measures such as $p$-values. We propose Matrix-Driven Instant Review (MDIR), leveraging matrix analysis and Large Deviation Theory. MDIR achieves accurate reconstruction of weight relationships, provides rigorous $p$-value estimation, and focuses exclusively on homologous weights without requiring full model inference. We demonstrate that MDIR reliably detects homology even after extensive mutations, such as random permutations and continual pretraining with trillions of tokens. Moreover, all detections can be performed on a single consumer PC, making MDIR efficient and accessible.

## 1 Introduction

Recent advances in large language models (LLMs) have led to widespread development and adaptation of models trained on massive datasets. While reusing model weights is generally harmless, issues arise when such reusage occurs without proper attribution to the original developers, especially in cases involving direct copying, upcycling (Yao et al., 2024; He et al., 2025), pruning (Ma et al., 2023; Meta-AI, 2024), or continual pretraining. The scale and complexity of LLMs make detection of model weight homology particularly challenging.

Existing methods for detecting model similarity can be broadly classified into two main categories: retrieval-based methods and representation-based methods.

**Retrieval-based Methods.** Retrieval-based Methods (Xu et al., 2024) rely on vendors embedding specific key-value pairs $\{(k_i, v_i)\}_{i=1,...,N}$ into training data, for instance, synthetic hexadecimal strings unlikely to occur naturally. During pretraining, models are optimized to maximize $p_\theta(v_i \mid k_i)$. Downstream models exhibiting anomalously high $p_{\theta'}(v_i \mid k_i)$ (or low perplexity) relative to a random baseline may then be flagged as derived from the original. This approach proved useful in a reported case involving Llama3-V and MiniCPM-o v2.6 (pzc163 et al., 2024), where rare oracle bone inscriptions were used as keys and their corresponding modern Chinese characters as values. However, its effectiveness depends on access to vendor-specific keys or prompts, which pose a significant practical constraint, as such data are rarely disclosed to external users.

**Representation-based Methods.** Representation-based methods such as REEF (Zhang et al., 2025), HuRef (Zeng et al., 2024), and Intrinsic Fingerprint (He et al., 2025) determine model similarity via comparing internal model representations (either weights directly, or activations under identical inputs). An ideal model similarity measure should be invariant to common transformations (e.g., permutation and scaling) and remain stable even after extensive continual pretraining. While these methods effectively reveal similarity through "fingerprints," they generally lack the ability to reconstruct the weight correspondence mapping (e.g., transformations such as neuron permutation or channel scaling) between models.

**Our Method.** We propose Matrix-Driven Instant Review (MDIR), a matrix-level similarity detection method. By directly operating on weight matrices, our method identifies not only similarity but also the transformations involved (including permutation and scaling) during model weight reusage.

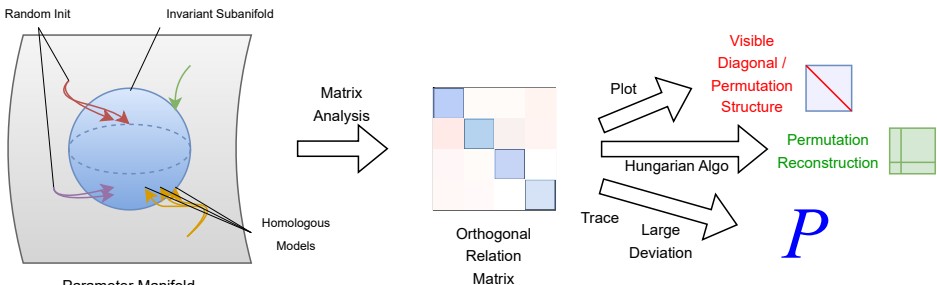

Figure 1: An overall illustration of our MDIR method.

Our method is grounded in a strong mathematical foundation, leveraging Singular Value Decomposition (SVD) and polar decomposition to analyze model matrices. We further utilize Large Deviation Theory (LDT) and random matrix theory to estimate $p$-values. These mathematical tools provide a rigorous framework for detecting similarities at a fundamental level, making it exceedingly difficult for adversaries to bypass our detection. Moreover, our method also democratizes the verification process. Without requiring access to vendor-specific prompts or specialized hardware, anyone with a standard PC can participate in the verification process. Our main process of MDIR can be summarized in Figure 1.

## 2 PRELIMINARIES

### 2.1 PROBLEM STATEMENT

In this paper, we focus solely on weight reuse of large language models and do not address similarities arising from training data selection. Specifically, given two LLMs $A$ and $B$, with their parameters denoted as $\theta_A$ and $\theta_B$, we aim to determine whether $A$ and $B$ exhibit a relationship in their weights, based solely on the statistical properties of $\theta_A$ and $\theta_B$. The weight relationships we consider include, but are not limited to, the following cases:

- **Fine-tuning**: Various kinds of SFT and RL included;
- **Continual Pretraining**: Training the model with more data in the general domain, sometimes as much as trillions of tokens;
- **Upcycling** (Yao et al., 2024; He et al., 2025): Continual pretraining with a larger model, especially an MoE model, with weights initialized from a smaller base model (usually dense);
- **Pruning** (Ma et al., 2023; Meta-AI, 2024): Removing certain channels or neurons of a base model to obtain a smaller model;
- **Transformation**: Should include permutations and even general orthogonal/unitary transformations;
- **Comprehensive Combination**: A combination of all these cases above.

This task can be formulated as a binary classification problem $\Psi$, where the inputs are $(\theta_A, \theta_B)$ and the output is $\Psi(\theta_A, \theta_B) \in \{0, 1\}$. Here, 1 indicates that the two models are homologous, while 0 indicates unrelated. Due to the vast number of parameters (on the order of billions) as problem input and the scarcity of examples, it is impractical to construct a reliable machine learning framework. Learning-based approach to this task would likely suffer from extreme overfitting without producing trustworthy signals.

### 2.2 MOTIVATION: INVARIANT TRANSFORMATIONS PRESERVE FUNCTIONALITY

Modern LLMs are massively overparameterized, meaning many distinct parameter configurations $\theta$ and $\theta'$ can produce identical input-output behavior: $f_\theta = f_{\theta'}$ even when $\theta \neq \theta'$. These function-

preserving configurations form what we call the **invariant space** of $\theta$:

$$\mathcal{M}_{\text{inv}}(\theta) = \{\theta' \in \mathcal{M}_{\text{arch}} \mid f_{\theta'} = f_\theta\},$$

where $\mathcal{M}_{\text{arch}}$ is the parameter space of the model architecture. However, the structure of the space $\mathcal{M}_{\text{inv}}(\theta)$ is often complicated. It may even vary in dimension (for example, its dimension can be higher when there are $n$ neurons sharing the same input weights in $\theta$).

We seek for a set of transformations, such as orthogonal transformations on the backbone and attention heads, that form a continuous group $G$ that acts on the weights without changing the model's behavior (i.e., the orbit is always generating a submanifold of $\mathcal{M}_{\text{inv}}(\theta)$). We call such transformations **totally invariant**:

**Definition 1.** *We call a Lie group $G$ acting on $\mathcal{M}_{arch}$ totally invariant if:*

1. *Every $g \in G$ preserves the input-output function: $f_{g\theta} = f_\theta$;*

2. *$G$ acts isometrically (preserves distances in parameter space).*

Note that, we do not need to characterize all totally invariant transformations; We need a subset with sufficiently large dimension, which is enough for subsequent analysis with high confidence. We construct such a group $G$ explicitly for Transformer architectures, especially with GQA, in Section 3.1.

**Key Insight: Training Trajectories Preserve $G$-Coordinates.** Under idealized conditions (infinite numerical precision and $G$-invariant optimizers), training dynamics are confined to the *quotient space* $\mathcal{M}_{\text{arch}}/G$. That is, the component of the parameters along the $G$-orbit (i.e., within the invariant group) remains unchanged throughout training.

To see why, consider a local orthogonal reparameterization near $\theta$: $(\alpha, \beta)$, where $\alpha$ parametrizes directions within $G\theta$ (the orbit), and $\beta$ parametrizes orthogonal directions that actually affect the function. Since the loss is invariant to $\alpha$, its gradient along $\alpha$ is zero:

$$\frac{\partial \text{Loss}}{\partial \alpha^{(i)}} = 0 \quad \forall i.$$

Thus, under idealized conditions, $\alpha$ remains constant at its initialization value.

If $G$ is an orthogonal subgroup, any orthogonal invariant optimizer is also $G$-invariant. This accounts for classical SGD and its momentum variants, but not the Adam(W) optimizer. In practice, however, it is unlikely that $\alpha$ deviates significantly from its original coordinates.

**Implication for Homology Detection.** Suppose two models $\theta_1, \theta_2$ are derived from the same initialization (e.g., one is fine-tuned from the other). Then their $G$-components $g_1, g_2$ should satisfy $g_1^{-1} g_2 \approx e$, where $e$ is the identity element of the group $G$. Intuitively, this means that the transformation aligning $\theta_1$ to $\theta_2$ lies close to the identity transformation. In contrast, independently initialized models will have $g_1^{-1} g_2$ distributed randomly across $G$. This suggests a simple homology detection criterion:

If $d_G(g_1^{-1} g_2, e)$ is small, then $\theta_1$ and $\theta_2$ are likely homologous,

where $d_G(\_, e)$ can simply be measured by Frobenius inner product, which is equal to the trace ($\langle g, e \rangle_F = \text{Tr}(g)$).

This principle directly enlightens our method MDIR: Using matrix decomposition techniques (SVD / polar), we compute $g_1^{-1} g_2$ and measure its deviation from the identity via the trace. When $g_1^{-1} g_2$ is close to identity, its pattern is clearly visible via the matrix plot. Subsequently, this structural pattern can be converted to statistical significance of $p$-value via Large Deviation Theory.

## 3 METHODOLOGIES

### 3.1 AN INVARIANT TRANSFORMATION GROUP FOR GQA

We assume that model $A$ is the original model and model $B$ the adversary, their parameters denoted as $\theta_A$ and $\theta_B$ respectively. Both models adopt a decoder-only Transformer architecture (Vaswani

et al., 2017; Radford et al., 2019), with word embeddings and unembeddings, Grouped Query Attention (GQA) (Ainslie et al., 2023), MLP layers with up and down projections (Shazeer, 2020), and RMSNorm layers (Zhang & Sennrich, 2019).

We select GQA for our analysis framework, as GQA represents the most prevalent form of attention mechanism in modern Transformers. Both Multi-Head Attention (with an expansion rate of 1) (Vaswani et al., 2017) and Multi-Query Attention (with one key-value head per layer) (Shazeer, 2019) can be treated as special cases of GQA.

**Transformations in the Attention Module.** For simplicity, we only consider one layer of attention, namely layer $\ell$. Assume that Model $B$ is equivalent to $A$ under certain transformations:

$$\theta_{A,\ell} = \{Q, K, V, O\}$$
$$\theta_{B,\ell} = \{Q', K', V', O'\}.$$

The linear weight transformation from $\theta_A$ to $\theta_B$ may take the following form:

$$Q' = U_Q Q W_Q, \quad K' = U_K K W_K, \quad V' = U_V V W_V, \quad O' = W_O^{-1} O U_O^{-1},$$

where $U_Q, U_K, U_V, U_O$ and $W_Q, W_K, W_V, W_O$ are transformation matrices applied to the original weights. These matrices represent modifications introduced during model adaptation.

We refer to $U_Q, U_K, U_V, U_O$ as *outer transformations*, which typically correspond to operations such as rotations, permutations, or scaled orthogonal transformations. Since both $U_Q, U_K, U_V$ and $Q, K, V$ operate on normalized vectors:

$$\text{RMSNorm}(x')U_\square = \text{RMSNorm}(x), \quad \square \in \{Q, K, V\},$$

this implies that $U_Q = U_K = U_V$, and they are all orthogonal matrices. In the context of Lie groups, we denote $U_Q = U_K = U_V \in \text{O}(\text{EmbDim})$.

For the *inner transformations* $W_Q, W_K, W_V, W_O$, the situation becomes more complex due to the presence of attention heads and nonlinear transformations (e.g., Softmax) across channels. Not all orthogonal transformations are permissible for inner transformations.

While we cannot prove that this set encompasses all possible transformations, we provide a sufficient subset:

$$W_Q = \mu \cdot P_1 \otimes P_2 \otimes S, \qquad W_K = \mu^{-1} \cdot P_1 \otimes S,$$
$$W_V = \sum_{v=1}^{\text{NumKVHeads}} (\mathbf{1}_{v,\sigma(v)} \otimes H_v), \quad W_O = \sum_{v=1}^{\text{NumKVHeads}} (\mathbf{1}_{v,\sigma(v)} \otimes P_2 \otimes H_v),$$

where $\otimes$ denotes the Kronecker product of matrices, $\mu \neq 0$ is a scalar, $P_1 \in \text{Perm}(\text{NumKVHeads})$ is a permutation matrix over $\text{NumKVHeads}$ channels (and $\sigma$ is the corresponding permutation such that $\sum \mathbf{1}_{v,\sigma(v)} = P_1$), $P_2 \in \text{Perm}(\text{QueriesPerHead})$ is a permutation matrix over $\text{QueriesPerHead}$ queries, $S \in \text{diag}(\pm 1, \cdots, \pm 1) \in \text{M}_{\text{HeadDim}}(\mathbb{R})$, and $H \in \text{O}(\text{HeadDim}, \mathbb{R})$ is an arbitrary orthogonal matrix.

To ensure compatibility with both QK-norm (Henry et al., 2020) and RoPE (Su et al., 2023), $S$ is restricted to diagonal matrices with entries $\pm 1$: $S \in \text{diag}(\pm 1, \cdots, \pm 1)$.

We summarize the GQA architecture and invariant transformations in Figure 2, illustrated using Penrose tensor notation, which compactly encodes the tensor contractions and symmetry operations underlying the GQA module.

**An Invariant Transformation Group.** Comprehensively, the *outer transformations* are directly attached to the model backbone channels (the place where the residual is added), and should be identical across all layers. The *inner transformations*, on the other hand, can be different for each layer. Let $E \in \mathbb{R}^{\text{VocabSize} \times \text{EmbDim}}$ denote the vocabulary embedding matrix (and $F$ the unembedding matrix) of model $A$, and $E' \in \mathbb{R}^{\text{VocabSize} \times \text{EmbDim}}$ the vocabulary embedding for model $B$

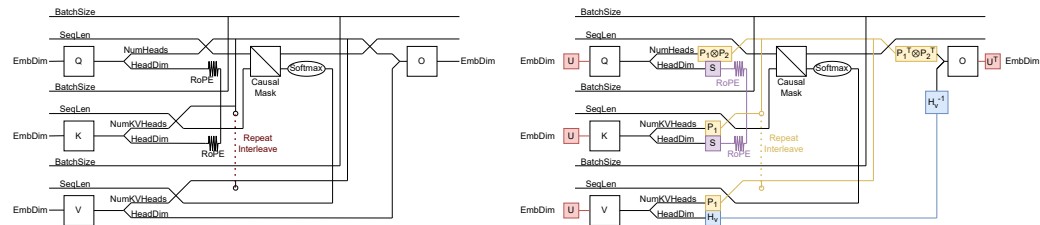

Figure 2: A Penrose notation of a model with Grouped Query Attention architecture (model $A$, left) and an illustration of an adversary $B$ under application of invariant transformations (right). Bias terms are omitted, and RMSNorms are not explicitly shown.

(and $F'$ the unembedding).

$$
\begin{aligned}
E' &= EU^{\mathrm{T}}, & F' &= UF, \\
Q'_\ell &= UQ_\ell W_{Q,\ell}, & K'_\ell &= UK_\ell W_{K,\ell}, \\
V'_\ell &= UV_\ell W_{V,\ell}, & O'_\ell &= W_{O,\ell}^{-1} O_\ell U^{\mathrm{T}}, \\
W_{Q,\ell} &= \mu_\ell \cdot P_{1,\ell} \otimes P_{2,\ell} \otimes S_\ell, & W_{K,\ell} &= \mu_\ell^{-1} \cdot P_{1,\ell} \otimes S_\ell, \\
W_{V,\ell} &= \sum_{v=1}^{\text{NumKVHeads}} (\mathbf{1}_{v,\sigma_\ell(v)} \otimes H_{\ell,v}), & W_{O,\ell} &= \sum_{v=1}^{\text{NumKVHeads}} (\mathbf{1}_{v,\sigma_\ell(v)} \otimes P_2 \otimes H_{\ell,v}), \\
(1 &\leq \ell \leq L)
\end{aligned}
$$

The group $G$ can be generated by these elements:

$$
U, \ (P_{1,\ell})_{1 \leq \ell \leq L}, \ (P_{2,\ell})_{1 \leq \ell \leq L}, \ (S_\ell)_{1 \leq \ell \leq L}, \ (H_{\ell,v})_{1 \leq \ell \leq L, 1 \leq v \leq \text{HeadDim}}.
$$

Whereas $U$ is the component that contributes the most dimension to this group.

### 3.2 Solving the Transformations

In an idealized setting, we have $E' = EU^{\mathrm{T}}$ for the vocabulary embeddings of models $A$ and $B$. However, in practical scenarios, the adversary might also have its model trained, perturbing the weight of $E'$. Following the established conventions, we have:

$$
E' = EU^{\mathrm{T}} + N_E,
$$

where $U \in \mathbb{R}^{\text{EmbDim} \times \text{EmbDim}}$ is an orthogonal matrix, and $N_E$ represents the additional perturbation introduced by training or noise injection.

To minimize the difference between $E'$ and $EX^{\mathrm{T}}$, we solve the following optimization problem:

$$
\min_{X \in \mathrm{O}(\text{EmbDim})} \|E' - EX^{\mathrm{T}}\|_F^2 = \min_{X \in \mathrm{O}(\text{EmbDim})} \left\langle E' - EX^{\mathrm{T}}, E' - EX^{\mathrm{T}} \right\rangle_F.
$$

Expanding the Frobenius norm yields:

$$
\begin{aligned}
\arg \min_{X \in \mathrm{O}(\text{EmbDim})} \|E' - EX^{\mathrm{T}}\|_F^2 &= \arg \min_{X \in \mathrm{O}(\text{EmbDim})} \left( \|E'\|_F^2 + \|E\|_F^2 - 2 \left\langle EX^{\mathrm{T}}, E' \right\rangle_F \right) \\
&= \arg \max_{X \in \mathrm{O}(\text{EmbDim})} \left\langle EX^{\mathrm{T}}, E' \right\rangle_F \\
&= \arg \max_{X \in \mathrm{O}(\text{EmbDim})} \mathrm{Tr} \left( EX^{\mathrm{T}} E'^{\mathrm{T}} \right) \\
&= \arg \max_{X \in \mathrm{O}(\text{EmbDim})} \mathrm{Tr} \left( (E'^{\mathrm{T}} E) X^{\mathrm{T}} \right).
\end{aligned}
$$

We denote $\tilde{U}$ as the solution to this optimization problem. From the trace maximization property, the solution of $\tilde{U}$ is equal to the orthogonal factor in the polar decomposition of $(E'^{\mathrm{T}} E)$. Note that $\tilde{U}$ is not the ground truth of $U$, but rather a close approximation. To reconstruct the actual $U$, we seek for

special structural patterns lying behind $\tilde{U}$. For example, if $\tilde{U}$ is sufficiently close to a permutation matrix $P \in \text{Perm}(\text{EmbDim})$, identifiable via:

$$P = \arg\max \text{Tr}(P\tilde{U}^{\mathrm{T}}),$$

then we may safely assert that $P$ is almost equal to the ground truth of $U$. This problem is equivalent to solving maximum bipartite matching or the linear sum assignment problem (SciPy, 2025), which can be computed in up to $O(n^3)$ time using the Hungarian algorithm. For large matrices (e.g., $18432 \times 18432$), solving this exactly may take longer on a CPU.

A unique determination of $\tilde{U}$ requires $(E'^{\mathrm{T}}E)$ to be non-degenerate and full-rank, which necessitates the vocabulary size to satisfy $\text{VocabSize} \geq \text{EmbDim}$. This condition is easily satisfied, as most recent tokenizers have $\text{VocabSize} \geq 3 \times 10^4$.

**Changed Tokenizer.** When model $B$ uses a different tokenizer, the embedding matrices $E$ and $E'$ are defined over different vocabularies. However, a substantial set of tokens $\mathcal{C}$, including ASCII bytes, common subwords (e.g., `is`, `take`), and morphemes (e.g., `-tion`), is typically shared. In contextualized representations, the meaning (and thus the embedding vector) of a token is determined by its usage across billions of contexts (Mikolov et al., 2013). Consequently, even after independent training, the embeddings of shared tokens in homologous models remain approximately aligned up to the global transformation $U$. Let $\mathcal{C}$ denote the set of all common tokens. We estimate $\tilde{U}$ as:

$$\tilde{U} = \arg\max_{X \in \mathrm{O}(\text{EmbDim})} \text{Tr}\left(\left(E'[\mathcal{C}, :]^{\mathrm{T}} E[\mathcal{C}, :]\right) X^{\mathrm{T}}\right).$$

Thus, $\tilde{U}$ corresponds to the orthogonal part in the polar decomposition of $(E'[\mathcal{C}, :]^{\mathrm{T}} E[\mathcal{C}, :])$.

### 3.3 ESTIMATING $p$-VALUE

After identifying a permutation matrix $P$ as $P = \arg\max \text{Tr}(P\tilde{U}^{\mathrm{T}})$. we need a statistical criterion to determine whether our identification is significant. While visual inspection of $P$ and $\tilde{U}$ can provide qualitative evidence, the value of $\text{Tr}(P\tilde{U}^{\mathrm{T}})$ itself serves as a strong quantitative indicator.

**Null Hypothesis.** Our null hypothesis assumes that models $A$ and $B$ are not homologous, and there is no apparent similarity between their weights. Specifically, we assume that $\tilde{U}$ is uniformly distributed over the orthogonal group $\mathrm{O}(n)$ according to the Haar measure. This assumption is reasonable because under the null hypothesis, there is no systematic relationship between the weights of $A$ and $B$, and any observed alignment would be purely coincidental.

Below, we take $n = \text{EmbDim}$ for short. Under the assumption of the null hypothesis, the probability measure $d\mathbb{P}$ should be uniform across all admissible transformations, and the distribution of $\tilde{U}$ should be uniform over $\mathrm{O}(n)$.

Now, fix $P_0$ as an arbitrary permutation matrix. We estimate the probability of $\text{Tr}(P_0\tilde{U}^{\mathrm{T}}) \geq c$, denoted as:

$$f(c) := \mathbb{P}\left[\text{Tr}(P_0\tilde{U}^{\mathrm{T}}) \geq c\right] = \mathbb{P}\left[\text{Tr}(\tilde{U}) \geq c\right],$$

since $P_0\tilde{U}^{\mathrm{T}}$ and $\tilde{U}$ are both uniformly distributed.

With $n!$ possible permutations, only the one maximizing $\text{Tr}(P\tilde{U}^{\mathrm{T}})$ is chosen. Applying the union bound over all $n!$ permutations yields

$$p \leq n! \cdot \mathbb{P}\left[\text{Tr}(P_0\tilde{U}^{\mathrm{T}}) \geq c\right] = n! \cdot f(c).$$

We estimate the $p$-value based on the evaluation of $f(c)$. Large Deviation Theory implies $f(c) \leq K \exp(-c^2/2)$ for some constant $K$. Thus, an upper bound is established as $\log p \leq \log(n!) - c^2/2 + \epsilon$. In practice, we set $\epsilon = 0$ for simplicity. Note that when $A$ and $B$ are homologous, $\text{Tr}(P_0\tilde{U}^{\mathrm{T}})$ scales linearly with $n$ (e.g., $c \approx 0.4n$), so the quadratic term $-c^2/2$ dominates $\log(n!)$, resulting in a highly significant $p$-value. Please refer to Appendix D for a detailed derivation.

# 4    EXPERIMENTS

## 4.1    OVERALL COMPARISON

We select 25 representative models for our comparison to evaluate the effectiveness of our MDIR method. For each pair of models, we compute the trace as $\max \mathrm{Tr}(P\tilde{U}^{\mathrm{T}})$ via $P = \mathtt{linear\_sum\_assignment}(\tilde{U})$ (SciPy, 2025). From the right part of Figure 3, we observe that

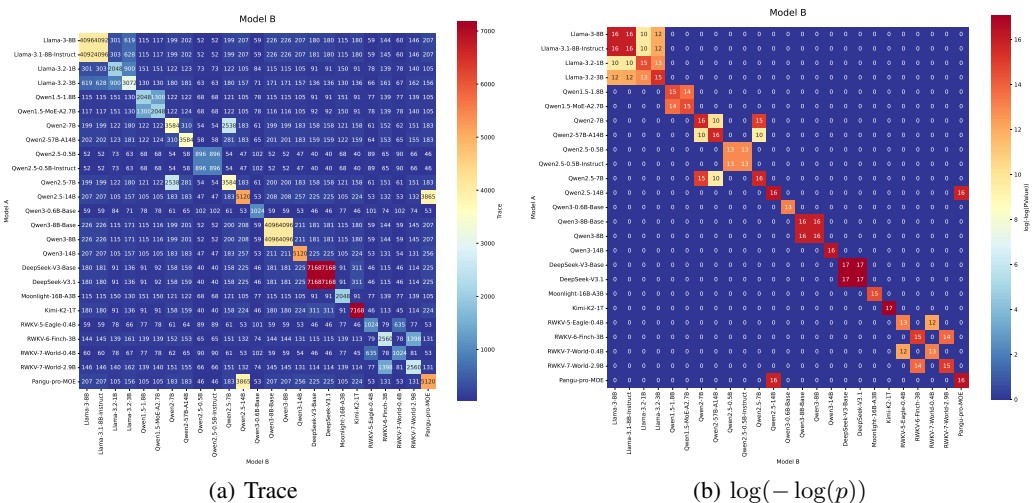

| (a) Trace | (b) $\log(-\log(p))$ |

Figure 3: Overall Comparison. We use $\log(-\log(p))$ to crop the $p$ values for better visualization. Value 0 in the right indicates no observable significance.

MDIR for homology detection is self-consistent — it fulfills the requirements of an equivalence relation: reflexivity, symmetry and transitivity. It has also correctly identified all known homology relations, including the following types:

- Instruction fine tuning and continual pretraining: Qwen2.5-0.5B-Instruct (Qwen et al., 2025), Llama-3.1-8B-Instruct (Meta-AI, 2024) and DeepSeek-V3.1 (DeepSeek-AI et al., 2025) are known to have trained from their predecessors;

- Pruning: Llama-3.2-1B and Llama-3.2-3B are pruned from Llama-3.1-8B (Meta-AI, 2024);

- Upcycling: Qwen1.5-MoE-A2.7B (Team) and Qwen2-57B-A14B (Yang et al., 2024) are upcycled from Qwen1.5-1.8B and Qwen2-7B, respectively;

- Non-transformer models: RWKV-7-World-0.4B and 2.9B (Peng et al., 2025) are upgraded from RWKV-5-Eagle-0.4B and RWKV-6-Finch-3B (Peng et al., 2024) respectively;

- Independently developed models: Moonlight-16B-A3B (Liu et al., 2025) and Kimi-K2 (Kimi-Team et al., 2025).

It is worth mentioning that our significance threshold is set purely *a priori*, based on the theoretical $p$-value bound, without any post-hoc calibration on known positive/negative pairs.

## 4.2    MATRIX VISUALIZATIONS

We visualize some matrices for several representative cases in Figures 4, 5 and 6.

## 4.3    ABLATION EXPERIMENT

To demonstrate that MDIR exclusively detects relevance in weights, rather than training data, we conducted an ablation experiment by initializing two models with different random seeds and train-

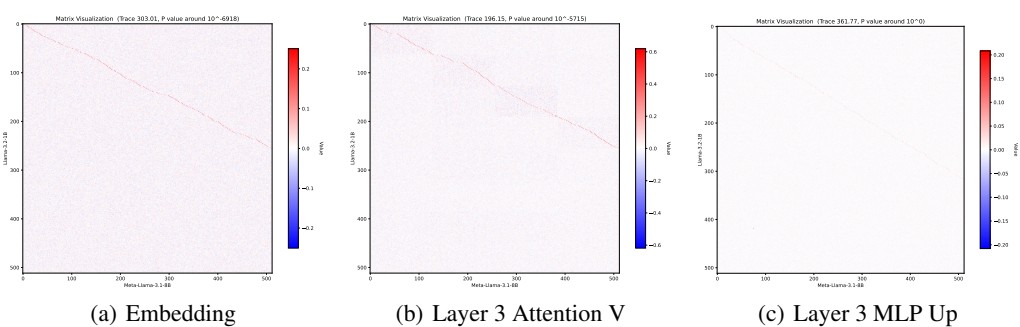

(a) Embedding        (b) Layer 3 Attention V        (c) Layer 3 MLP Up

Figure 4: MDIR suggests homology between Llama-3.1-8B and Llama-3.2-1B. yielding a $p$-value of $10^{-6,918}$. For model pruning, the irregular oblique curves (the slope is approximately $1/2$, indicating that half of the channels are retained) can be clearly identified in $\tilde{U}$ from vocabulary as well as inner transformations in the attention module.

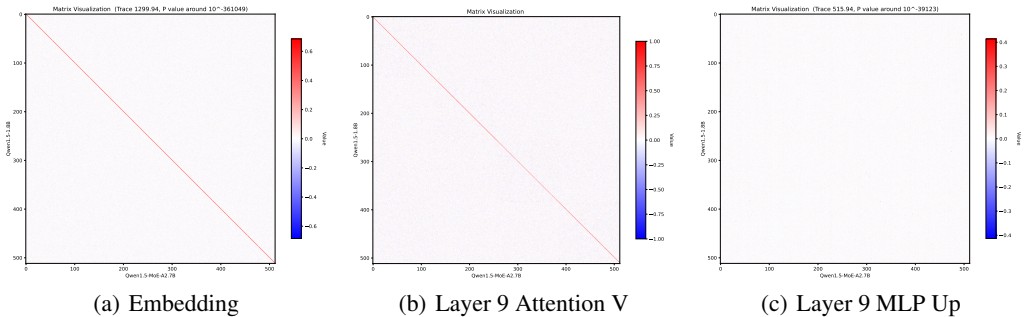

(a) Embedding        (b) Layer 9 Attention V        (c) Layer 9 MLP Up

Figure 5: For model upcycling, MDIR suggests homology between Qwen1.5-1.8B and Qwen1.5-MoE-A2.7B, yielding a $p$-value of $10^{-361,049}$. The diagonal patterns for vocabulary embedding and attention modules indicate that these modules are directly inherited from its predecessor, and show no evidence of permutation or channel reselection before the upscaling process.

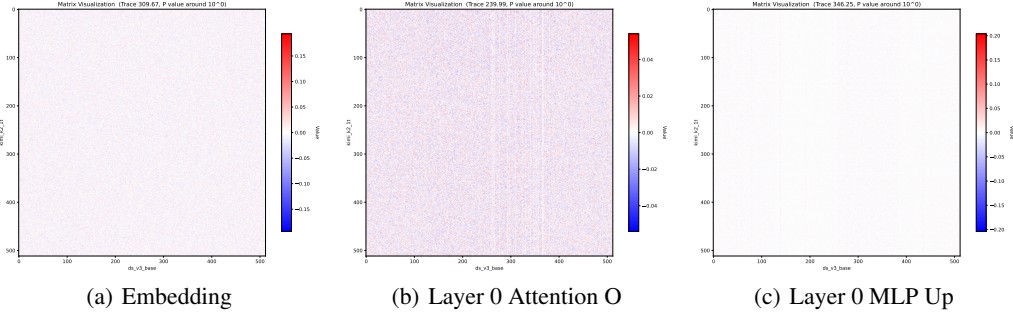

(a) Embedding        (b) Layer 0 Attention O        (c) Layer 0 MLP Up

Figure 6: For independently developed models, MDIR detects no statistically significant homology between DeepSeek-V3-Base and Kimi-K2-Instruct, with no clear pattern or statistically significant $p$-value observed.

ing each on two distinct datasets, resulting in a total of 4 models. The datasets were DCLM sub-sample (Li et al., 2025) (the first 100 files of shard 0 (MLFoundations, 2024), 12.05 billion tokens), and OpenWebMath-ProX (Zhou et al., 2025) (4.61 billion tokens (Gair-ProX, 2024)). Both datasets were tokenized using the GPT-NeoX tokenizer (Black et al., 2022). Models were configured with the `Qwen3ForCausalLM` (HuggingFace, 2025) architecture, with 12 layers and an intermediate size of 1024, resulting in a total of 291 million parameters. They were initialized using HuggingFace `transformers`' default initialization range of 0.02, with random seeds 2 and 3, respectively. All models were trained with a learning rate of linear warmup to 0.002 followed by quadratic decay to 0, and batch size $8(\text{GPUs}) \times 48(\text{sequences}) \times 1024(\text{length})$.

The left two subfigues of Figure 7 reveal clear diagonal patterns for models initialized with the same seed, indicating strong weight similarity due to shared initialization. In contrast, the right two subfigures show no significant outliers and no significance of $p$-values, even though block-wise patterns are present for inner transformation matrices of attention module. This suggests that models trained on same dataset may develop similar attention features but no substantial weight correlation.

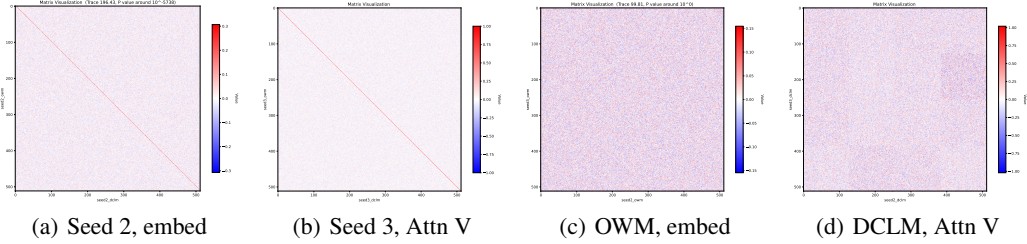

| (a) Seed 2, embed | (b) Seed 3, Attn V | (c) OWM, embed | (d) DCLM, Attn V |

Figure 7: MDIR only identifies relationship between models initialized with the same seed.

## 5 CONCLUSION

We introduced MDIR, a novel method for detecting weight homology in large language models, leveraging matrix analysis and Large Deviation Theory to provide statistically rigorous results. MDIR operates directly on model weights (bypassing full inference) and runs efficiently on a consumer PC, democratizing the verification process. Crucially, MDIR detects weight-level reuse rather than data-level similarity, making it more specific and targeted than previous techniques. This reduces the likelihood of false positives and enhances reliability.

**Why Extreme $p$-values.** Our method produces $p$-values as small as $10^{-10^4}$ or lower, far beyond typical statistical thresholds. This is not a numerical artifact, but a direct consequence of Large Deviation Theory (LDT). LDT characterizes tail probabilities via rate functions: $p \asymp \exp(-n^2 I(x))$, where $n$ is the model dimension and $I$ is some rate function quantifying deviation from the null. For modern LLMs with $n \gg 1$, this could lead to astronomically small $p$-values. This reflects the fundamental difference between LLM-scale statistics (billions of parameters) and classical statistical scenarios (hundreds of samples). Such values underflow in floating-point arithmetic but remain well-defined and computable via $\log p$.

**Limitation: Numerical Precision.** Our analysis assumes infinite numerical precision. In practice, polar decomposition becomes unstable for near-low-rank matrices, and training in low-precision formats (fp16/bf16/fp8) may perturb the Haar-measure assumption. We observed up to 1% discrepancy in the polar factor when switching between fp32 and fp64—though this did not affect detection outcomes in our experiments.

**Future Work: Potential Ways of Evading Detection.** Although not observed in our experiments, there might be methods to evade our detection. Could MDIR be evaded via more complex adversaries, like additional training with larger learning rates, especially for vocabulary embeddings and attention modules? We leave this for future work.

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

# A    MATHEMATICAL BACKGROUND

Our work primarily builds upon the foundation of matrix analysis. To avoid excessive technicality or verbosity, we only introduce the key tools and properties used in this paper, potentially omitting or abbreviating proofs. Interested readers are encouraged to refer to Horn & Johnson (2012) for a comprehensive overview of matrix analysis.

## A.1    MATRIX ANALYSIS: SINGULAR VALUE AND POLAR DECOMPOSITIONS

**Singular Value Decomposition.** The singular value decomposition (SVD) of a matrix $A \in \mathbb{R}^{m \times n}$ is given by

$$A = USV^{\mathrm{T}},$$

where $U \in \mathbb{R}^{m \times m}$ and $V \in \mathbb{R}^{n \times n}$ are orthogonal matrices, and $S \in \mathbb{R}^{m \times n}$ is a diagonal matrix with non-negative singular values on the diagonal:

$$\sigma_1 \geq \sigma_2 \geq \cdots \geq \sigma_r > 0 \quad (r = \mathrm{rank}(A)).$$

We denote $\sigma_i(A)$ as the $i$-th singular value of $A$.

**Polar Decomposition.** The polar decomposition of $A$ has two common but distinct forms: the left decomposition $A = PW$ and the right decomposition $A = WQ$, where $P = (AA^{\mathrm{T}})^{1/2}$ and $Q = (A^{\mathrm{T}}A)^{1/2}$ are symmetric positive semidefinite matrices, and $W \in \mathbb{R}^{m \times n}$ (with orthonormal columns) is shared between both decompositions. We define the *orthogonal part* of $A$ as $\mathrm{Ortho}(A) := W$. When $A$ is full-rank, the orthogonal part of $A$ is unique.

**Connection to SVD.** The orthogonal part $W$ in the polar decomposition can be obtained from the SVD:

$$W = UV^{\mathrm{T}},$$

where $U$ and $V$ are derived from the SVD of $A$. The symmetric factors satisfy $P = USU^{\mathrm{T}}$ and $Q = VSV^{\mathrm{T}}$. However, when $A$ is not invertible (with many singular values either exactly zero or close to zero), the computation of $W = UV^{\mathrm{T}}$ becomes ill-conditioned.

**SVD and Spectral Calculus.** Assume $A \in \mathbb{R}^{m \times n}$ has SVD $A = USV^{\mathrm{T}}$. Then:

$$AA^{\mathrm{T}} = U(SS^{\mathrm{T}})U^{\mathrm{T}}, \quad A^{\mathrm{T}}A = V(S^{\mathrm{T}}S)V^{\mathrm{T}}.$$

For any polynomial function $f \in \mathbb{R}[x]$ and $g(x) = xf(x^2)$, we have:

$$
\begin{aligned}
&f(AA^{\mathrm{T}})A \\
&= Uf(SS^{\mathrm{T}})U^{\mathrm{T}}USV^{\mathrm{T}} = Uf(SS^{\mathrm{T}})SV^{\mathrm{T}} \\
&= Ug(S)V^{\mathrm{T}} \\
&= USf(S^{\mathrm{T}}S)V^{\mathrm{T}} = USV^{\mathrm{T}}Vf(S^{\mathrm{T}}S)V^{\mathrm{T}} \\
&= Af(A^{\mathrm{T}}A).
\end{aligned}
$$

In fact, $g(S)$ can represent any odd polynomial. For any odd function $G$, we may find a polynomial $f$ such that $f(x)$ agrees with $G(x)/x$ on all nonzero diagonal entries of $SS^{\mathrm{T}}$. In particular, setting $G(x) = \mathrm{sign}(x)$, this spectral calculus yields the most "sensible" orthogonal part of $A$. We adopt the methods from Amsel et al. (2025) for a fast and relatively accurate implementation.

**Trace Maximization Property.** The orthogonal part $W$ solves the optimization problem:

$$\max_{\{X | X^{\mathrm{T}}X = I\}} \left( \mathrm{Tr}(AX^{\mathrm{T}}) \right).$$

The maximum value is the sum of all the singular values of $A$, i.e., $\mathrm{Tr}(P) = \sum_{i=1}^{r} \sigma_i$.

**Orthogonal Invariance of Singular Values.** The singular values $\{\sigma_i\}$ of $A$ are invariant under both left and right orthogonal transformations:

$$\sigma_i(UAV) = \sigma_i(A), \quad \forall U, V \text{ orthogonal and } i = 1, \cdots, \min(m, n).$$

Thus, any function depending on these singular values remains invariant under orthogonal transformations. Examples include:

- **Frobenius Norm**: $\|A\|_F = \sqrt{\sum_i \sigma_i^2(A)}$;

- **Spectral Norm**: $\|A\|_S = \sigma_1(A)$;

- **Ky Fan $k$-Norm**: $\|A\|_{KF} = \sum_{i=1}^k \sigma_i(A)$;

- **Schatten $p$-Norm** (Schatten, 1950): $\|A\|_p = \sqrt[p]{\sum_i \sigma_i^p(A)}$.

Any combination of these functions also remains invariant under orthogonal transformations. These functions may serve as preliminary indicators for detecting model similarity.

**Orthogonal Matrices and RMSNorms Commute.** For any orthogonal matrix $U$, we have:

$$\text{RMSNorm}(x)U = \text{RMSNorm}(xU),$$

for any nonzero vector $x \in \mathbb{R}^n$ (row vector). Moreover, all transformations satisfying this property $(\text{RMSNorm}(\cdot))U = \text{RMSNorm}((\cdot)U)$ are orthogonal transformations.

To prove this, note that $\text{RMSNorm}(y)$ is always a constant multiple $(\sqrt{n})$ of a unit vector when $y \neq 0$. Thus, $U$ maps all unit vectors to unit vectors. By linearity, this implies $\|xU\| = \|x\|$ and $xUU^\mathsf{T}x^\mathsf{T} = xx^\mathsf{T}$. Taking $x$ over all eigenspaces of $UU^\mathsf{T}$, we see that $1$ is the only eigenvalue of $UU^\mathsf{T}$. Hence, $UU^\mathsf{T} = \mathbf{1}_n$, and $U$ is orthogonal.

## A.2 LARGE DEVIATION THEORY

Our research extensively involves random orthogonal matrices, particularly focusing on traces of such matrices. To obtain statistically meaningful $p$-values, traditional statistical $p$-tests become useless for our case, due to the interdependence of the elements within an orthogonal matrix. Instead, we rely on large deviation theory for deriving the $p$-value. For a comprehensive exposition of this theory, please refer to Dembo & Zeitouni (1998) and Anderson et al. (2010).

# B SOLVING ALL TRANSFORMATIONS

If the significance of the $p$-value has already been determined from the $p$-values for vocabulary embeddings, this step is entirely optional. However, if the no significant $p$-value is observed at this stage, we cannot yet rule out the possibility of model homology. This may occur when an adversary used a general orthogonal matrix for obfuscation. Additionally, interested readers may wish to determine the exact relationship between two models, potentially with different architectures.

Since $P = \arg\max \text{Tr}(P\tilde{U}^\mathsf{T})$, we proceed as follows: If $P$ is reliably identified as a permutation matrix ($p < 2 \times 10^{-23}$), set $U := P$; otherwise, set $U := \tilde{U}$.

**Solving the Relationship between Layers.** In practical scenarios, the number of layers for two models are not necessarily the same. It is important to first determine the relationship between the layers of both models via solving a maximal ordered matching problem. We select a representative matrix (for example, attention K or V) for each layer $i$ of model $A$ and layer $j$ of model $B$. For each pair $(i, j)$ we do the process below and obtain a similarity measure matrix of shape $L \times L'$. Solving the maximal ordered matching for this matrix will give the relationship between all layers. See Section E.1 for a specefic example how this works.

**Solving the Transformations in the Attention Module.** We now solve for the inner transformations $W_Q$, $W_K$, $W_V$, and $W_O$ based on the heuristic $U$.

Our objectives are:

$$\min_{W_Q \text{ orthogonal}} \|UQW_Q - Q'\|_F^2, \quad \min_{W_K \text{ orthogonal}} \|UKW_K - K'\|_F^2,$$

for $Q$ and $K$. By the trace maximization property, the solutions for $W_Q$ and $W_K$ are given by:

$$W_Q = \lambda \text{Ortho}(Q^\mathsf{T}U^\mathsf{T}Q'), \quad W_K = \mu \text{Ortho}(K^\mathsf{T}U^\mathsf{T}K'),$$

where the similarity measure can be calculated as

$$t_\square = \texttt{linear\_sum\_assignment}(W_\square), \quad \square \in Q, K.$$

Also, scaling coefficients $\lambda$ and $\mu$ are computed as:

$$\lambda = \frac{\|Q'\|_F}{\|Q\|_F}, \quad \mu = \frac{\|K'\|_F}{\|K\|_F}.$$

For $W_V$ and $W_O$, we may apply the same method to compute $\mathrm{Ortho}(V^\mathrm{T}U^\mathrm{T}V')$ and $\mathrm{Ortho}(OU^\mathrm{T}O'^\mathrm{T})$. However, since $W_V$ and $W_O$ may involve general invertible transformations, our method does not guarantee recovering the exact transformation. solving the general case without additional assumptions is challenging. We leave this problem for future work.

**Solving the MLP.** At this point, there is only one matrix left to solve: the permutation of intermediate neurons, denoted by $P$. The solution is given as follows:

$$U_X = \mathrm{Ortho}(X^\mathrm{T}U^\mathrm{T}X'), \quad X \in \{\mathrm{Gate}, \mathrm{Up}, \mathrm{Down}\};$$

$$P = \arg \max_{P \in \mathrm{Perm(IntermediateDim)}} \mathrm{Tr}\left(P\left(U_\mathrm{Gate} + U_\mathrm{Up} + U_\mathrm{Down}\right)^\mathrm{T}\right).$$

Typically, we expect the three solutions $\arg \max \mathrm{Tr}\left(PU_X^\mathrm{T}\right)$, for $X \in \{\mathrm{Gate}, \mathrm{Up}, \mathrm{Down}\}$, to yield the same permutation. However, computing the orthonormal part for intermediate matrices (which often have 14,000–20,000 rows) is computationally expensive. Adding the three terms together would triple the computation cost. If the noise level is tolerable, we may simply select one of them:

$$P = \arg \max_{P \in \mathrm{Perm(IntermediateDim)}} \mathrm{Tr}\left(PU_\mathrm{Up}^\mathrm{T}\right).$$

## C  OVERALL ALGORITHM

We summarize our algorithm as follows:

---
**Algorithm 1** Computing the $p$-value: $\mathrm{PValue}(P, U, d)$

---
**Input:** Permutation matrix $P$, orthogonal matrix $U$, dimension $d$
Compute $p := d! \cdot \exp(-(\mathrm{Tr}(PU^\mathrm{T}))^2/2)$ via $\log(p) \approx (d \log d - d) - (\mathrm{Tr}(PU^\mathrm{T}))^2/2$
**Output:** $p$

---

## D  ESTIMATION OF $p$-VALUE VIA LARGE DEVIATION THEORY

Let $\tilde{U}$ be a random orthogonal matrix, distributed uniformly according to the normalized Haar measure. We aim to estimate the following function, especially its long-tail behavior, for $c > 0$:

$$f(c) = \mathbb{P}\left[\mathrm{Tr}(\tilde{U}) \geq c\right].$$

The distribution of $f(c)$ is a well-studied problem in random matrix theory. Diaconis & Shahshahani (1994) proved that $\mathrm{Tr}(\tilde{U}) \to \mathcal{N}(0, 1)$ in distribution. Later, Johansson (1997) showed that the convergence of $\mathrm{Tr}(\tilde{U})$ to $\mathcal{N}(0, 1)$ is exponential under the total variation distance:

$$\mathrm{TV}(f - (1 - \Phi)) \leq \exp(-\alpha n), \quad \text{for some } \alpha > 0,$$

where $\Phi(x)$ is the cumulative distribution function of the standard normal distribution.

It is known that $(1 - \Phi)(x)$ has the following asymptotic behavior for large $x$:

$$(1 - \Phi)(x) \approx \frac{1}{\sqrt{2\pi}x} \exp\left(-\frac{x^2}{2}\right), \quad (x \gg 1).$$

However, when both $n$ and $x$ are large, $\exp(-\alpha n)$ remains significantly larger than $\frac{1}{\sqrt{2\pi}x} \exp\left(-\frac{x^2}{2}\right)$. Thus, it is inappropriate to use $(1 - \Phi)$ as the asymptotics of $f(c)$.

To study the tail behavior of $f(c)$, we leverage tools from Large Deviation Theory. Since $n$ is typically large and the embedding dimension $n$ is usually an even number in most models, we

---

**Algorithm 2** MDIR (Matrix-Driven Instant Review)

---

**Input:** Model $A$, Model $B$

**Initialize:** Threshold $p_0 =$ some threshold (e.g. $\Phi(-10)$), relation flag $r = \mathsf{False}$

$L :=$ Number of layers in the model

$E :=$ Embedding matrix of $A$

$E' :=$ Embedding matrix of $B$

$\mathcal{C} :=$ Set of common vocabulary tokens between $A$ and $B$

Compute $\tilde{U} := \mathrm{Ortho}(E[\mathcal{C},:]^{\mathsf{T}} E'[\mathcal{C},:])$ via polar decomposition

$P := \arg\max_{P \in \mathrm{Perm(EmbDim)}} \mathrm{Tr}(P\tilde{U}^{\mathsf{T}})$

Compute $p := \mathrm{PValue}(P, \tilde{U}, \mathrm{EmbDim})$

**if** $p < p_0$ **then**

    Set $r := \mathsf{True}$

    Set $U := P$

**else**

    Set $U := \tilde{U}$

**end if**

**Yield:** Transformation matrix $U$, $p$-value $p$

**for** Layer $i \in [L]$ **do**

    Extract attention weights $Q, K, V, O$ from layer $i$ of $A$

    Extract attention weights $Q', K', V', O'$ from layer $i$ of $B$

    **for** $X \in \{Q, K, V\}$ **do**

        Compute $W_X := \mathrm{Ortho}(X^{\mathsf{T}} U^{\mathsf{T}} X') \cdot \frac{\|X'\|_F}{\|X\|_F}$

    **end for**

    Extract MLP weights $\mathrm{Gate}, \mathrm{Up}, \mathrm{Down}$ from layer $i$ of $A$

    Extract MLP weights $\mathrm{Gate}', \mathrm{Up}', \mathrm{Down}'$ from layer $i$ of $B$

    Compute $U_X := \mathrm{Ortho}(X^{\mathsf{T}} U^{\mathsf{T}} X')$ for $X \in \{\mathrm{Up}\}$ (or $X \in \{\mathrm{Gate}, \mathrm{Up}, \mathrm{Down}\}$)

    $P := \arg\max_{P \in \mathrm{Perm(IntermediateDim)}} \mathrm{Tr}(P U_{\mathrm{Up}}^{\mathsf{T}})$

**end for**

**Output:** Relation flag $r$

---

assume $n = 2m$ without loss of generality. Additionally, we assume $\det(\tilde{U}) = 1$, or equivalently $\tilde{U} \in \mathrm{SO}(2m)$, since this introduces only an infinitesimal difference in the thermodynamic limit $(n \to \infty)$. These assumptions simplify our analysis while preserving accuracy.

**Theorem 1.** *Let $A \in \mathrm{SO}(2m)$ be uniformly distributed according to the normalized Haar probability measure, and let $0 < r \leq 1/2$. The probability*

$$P(r, m) = \mathbb{P}\left[\frac{1}{2m}\mathrm{Tr}(A) \geq r\right]$$

*satisfies the following large deviation principle:*

$$\lim_{m \to \infty} \frac{-\log P(r, m)}{2m^2 r^2} = 1,$$

*or equivalently,*

$$P(r, m) \asymp \exp(-m^2 I(r)),$$

*where*

$$I(r) = 2r^2$$

*is the good rate function.*

*For $r > 1/2$, a faster decay rate can be achieved: $I(r) > 2r^2$.*

Before we delve into the proof, it is worth discussing the challenges we encountered. We are dealing with the Circular Real Ensemble (CRE, named after Serban et al. (2010)), which has a formulation distinct from traditional circular ensembles, such as Circular Unitary Ensembles (CUE) or Circular Orthogonal Ensembles (COE) (Dyson, 1962). These ensembles play important roles in both random matrix theory and condensed matter physics.

One of the problems most similar to ours is the Gross-Witten Ensemble (Gross & Witten, 1980; Gamboa et al., 2017), which concerns the large deviations from the typical value of $\mathrm{Re}\,\mathrm{Tr}(U)$ as $N \to \infty$ (where $U \in \mathrm{U}(N)$ is a random unitary matrix). Our problem can be regarded as a real orthogonal variant of Gross-Witten Ensemble, but this is undocumented in previous literature.

Unitary matrices possess many elegant properties. One of them is the Cayley transformation, defined as $\phi(z) := \mathrm{i}(1 + z)/(1 - z)$, which maps the punctured unit circle $\mathbb{T} \setminus \{1\}$ to the real line $\mathbb{R}$. When the Cayley transformation is applied to a unitary matrix, it sends unit eigenvalues to the real line, thereby producing a Hermitian matrix (i.e., $\phi(U)$ is Hermitian for $U \in \mathrm{U}(N)$). However, under the Cayley transformation, an orthogonal matrix is transformed into a purely imaginary, anti-symmetric matrix, which is of little interest. In the field of deep learning, complex values rarely appear, and we shall focus on the Circular Real Ensemble in our proof.

The circular ensemble is closely related to thermodynamics and large deviation theory. We refer to Mehta's book *Random Matrices* (Mehta, 2004), and a comprehensive introduction can be found in Chapter 12 of it.

### D.1 PROOF OF THEOREM 1

*Proof.* If $A$ is uniformly distributed according to the Haar measure in $\mathrm{SO}(2m)$ (a manifold of real dimension $m(2m - 1)$), all eigenvalues of $A$ lie on the unit circle, and complex eigenvalues form paired conjugates, possibly accompanied by several $+1$ and $-1$.

When $\det A = 1$, the product of complex eigenvalues yields $+1$, and there are almost surely no $-1$ or $+1$ eigenvalues.

Denote by $\{e^{i\theta_k}, e^{-i\theta_k} : k = 1, 2, \cdots, m\}$ the eigenvalues of $A$. It is a classical result (see Weyl (2016); Girko (1985)) that the phases $(\theta_k)_k$ obey the distribution characterized by the following probability density:

$$p(\theta_1, \cdots, \theta_m)\mathrm{d}\theta_1 \cdots \mathrm{d}\theta_m = C \prod_{1 \leq k < j \leq m} (\cos\theta_k - \cos\theta_j)^2 \mathrm{d}\theta_1 \cdots \mathrm{d}\theta_m,$$

and the trace $\mathrm{Tr}(A)$ is the sum of all eigenvalues:

$$\mathrm{Tr}(A) = 2\sum_{i=1}^{m} \cos(\theta_i).$$

By substitution of variables, let $t_i = \cos(\theta_i) \in [-1, 1]$, and $\mathrm{d}t_i/\sqrt{1 - t_i^2} = \mathrm{d}\theta_i$. We study the substituted distribution:

$$p(t_1, \cdots, t_m)\mathrm{d}t_1 \cdots \mathrm{d}t_m = C' \prod_{1 \le k < j \le m} (t_k - t_j)^2 \cdot \prod_{1 \le i \le m} (1 - t_i^2)^{-1/2}\mathrm{d}t_1 \cdots \mathrm{d}t_m.$$

Taking the logarithm, we have

$$-\log p(t_1, \cdots, t_m) = \sum_{1 \le k < j \le m} (-2\log|t_k - t_j|) + \sum_{1 \le i \le m} \left( \frac{\log(1 - t_i^2)}{2} \right) + C_0.$$

This has a clear thermodynamical interpretation: Consider $m$ interacting particles located on the interval $[-1, 1]$, with coordinates $t_1, \cdots, t_m$. The energy $E(t_1, \cdots, t_m)$ is the sum of the following two kinds of potentials:

1. Repelling force: for particles $k$ and $j$, their interaction potential is $-2\log|t_k - t_j|$, meaning that they repel each other according to the 2-dimensional Coulomb law;

2. External field: particles are attracted to the boundary points $-1$ and $+1$, with the potential $\sum_{1 \le i \le m} \left( \frac{\log(1 - t_i^2)}{2} \right)$.

Inspired by Chapter 12 of Mehta (2004) and also Touchette (2009), we define the Canonical Ensemble of these $m$ particles as follows, which admits a partition function:

$$Z(t_1, \cdots, t_m) = \int_{-1}^{1} \cdots \int_{-1}^{1} \exp(-\beta E(t_1, \cdots, t_m))\mathrm{d}t_1 \cdots \mathrm{d}t_m,$$

where

$$E(t_1, \cdots, t_m) = \sum_{1 \le k < j \le m} (-2\log|t_k - t_j|) + \sum_{1 \le i \le m} \left( \frac{\log(1 - t_i^2)}{2} \right)$$
$$= -\log p(t_1, \cdots, t_m) - C_0.$$

We set the thermodynamic beta to be $\beta = 1/(k_B T) = 1$ because we do not study temperature changes.

We are interested in the probability:

$$P(r, m) = \mathbb{P}\left[ \frac{1}{2m} \operatorname{Tr}(A) \ge r \right].$$

Using the representation $\operatorname{Tr}(A) = 2\sum_{k=1}^{m} \cos(\theta_k) = 2\sum_{k=1}^{m} t_k$, this becomes:

$$P(r, m) = \mathbb{P}\left[ \frac{1}{m} \sum_{k=1}^{m} t_k \ge r \right].$$

Let

$$\mu_m = \frac{1}{m} \sum_{k=1}^{m} \delta_{t_k}$$

be the empirical measure of the eigenvalue distribution. In the thermodynamic limit $m \to \infty$, $\mu_m$ should converge weakly to an equilibrium measure $\mu$ that minimizes the free energy functional $F(\mu)$.

We now solve the exact form of $F(\mu)$. We refer to Theorem 2.1 of the paper (Eichelsbacher et al., 2011), where we are dealing with the case $\theta = 1$, $\kappa = 1$, and $w_m(x) = (1 - x^2)^{1/(2m)}$, where $w_m(x) \to 1$ in the thermodynamic limit $m \to \infty$. This suggests that the interaction term dominates, and the external field term is negligible when $m$ is large.

The form of $F(\mu)$ is therefore given by

$$F(\mu) = \iint_{[-1,1]^2} \left( \log \frac{1}{|x - y|} \right) \mathrm{d}\mu(x)\mathrm{d}\mu(y),$$

with the rate function

$$\tilde{I}(\mu) = \iint_{[-1,1]^2} \left( \log \frac{1}{|x-y|} \right) \mathrm{d}\mu(x)\mathrm{d}\mu(y) - c,$$

where $c = \inf_\mu F(\mu)$. Note that the rate function $\tilde{I}(\mu)$ is defined for the probability measure $\mu$, not yet ready for our $I(r)$.

From Corollary 2.2 of the paper (Eichelsbacher et al., 2011), it suffices to solve the following two variational problems:

$$\inf_\mu F(\mu); \qquad \inf_{\mu:\, \int x\mathrm{d}\mu \geq r} F(\mu).$$

To solve these two problems, we parametrize $\mu$ using Chebyshev polynomials, with the additional assumption that $\mu$ is absolutely continuous with respect to the Lebesgue measure $\mathrm{d}x$ and normalized Chebyshev measure $\mathrm{d}x/(\pi\sqrt{1-x^2})$.

We suppose that $\mu$ is parametrized by the following series:

$$\mu(x) = \frac{1}{\pi\sqrt{1-x^2}} \sum_{i=0}^{\infty} a_i T_i(x),$$

where $T_i$ is the $i$-th Chebyshev polynomial of the first kind, and $F(\mu)$ now becomes a quadratic form of these coefficients $\{a_i\}$. An additional constraint must not be overlooked: $\mu$ is a probability measure, and

$$\int_{-1}^{1} \mathrm{d}\mu = \frac{T_0(x)}{\pi\sqrt{1-x^2}} \sum_{i=0}^{\infty} a_i T_i(x)\mathrm{d}x = a_0 = 1.$$

It is known that $\log|x-y|$ has the Chebyshev expansion (Sloan & Stephan, 1992; Mason & Handscomb, 2002):

$$\log|x-y| = -\log 2 - \sum_{n=1}^{\infty} \frac{2}{n} T_n(x)T_n(y),$$

so

$$F(\mu) = \iint_{[-1,1]^2} \left( \log 2 + \sum_{n=1}^{\infty} \frac{2}{n} T_n(x)T_n(y) \right) \frac{1}{\pi^2\sqrt{1-x^2}\sqrt{1-y^2}}$$

$$\left( \sum_{i=0}^{\infty} a_i T_i(x) \right) \left( \sum_{i=0}^{\infty} a_i T_i(y) \right) \mathrm{d}x\mathrm{d}y$$

$$= \log 2 \cdot \iint_{[-1,1]^2} \frac{1}{\pi^2\sqrt{1-x^2}\sqrt{1-y^2}} \mathrm{d}x\mathrm{d}y$$

$$+ \iint_{[-1,1]^2} \left( \sum_{n=1}^{\infty} \frac{2}{n} T_n(x)T_n(y) \right) \frac{1}{\pi^2\sqrt{1-x^2}\sqrt{1-y^2}}$$

$$\left( \sum_{i=1}^{\infty} a_i T_i(x) \right) \left( \sum_{i=1}^{\infty} a_i T_i(y) \right) \mathrm{d}x\mathrm{d}y.$$

We recall the orthogonal relation of Chebyshev polynomials,

$$\int_{-1}^{1} \frac{T_a(x)T_b(x)}{\pi\sqrt{1-x^2}}\mathrm{d}x = \begin{cases} 1, & a = b = 0; \\ \dfrac{1}{2}, & a = b \neq 0; \\ 0, & a \neq b. \end{cases}$$

Using the orthogonal relation, we further simplify $F(\mu)$ as

$$F(\mu) = \log 2 + \sum_{i=1}^{\infty} \frac{a_i^2}{2i}.$$

Therefore, the problem $\inf_\mu F(\mu)$ admits a simple solution: $a_i = 0$ for every $i \geq 1$, and

$$d\mu_0 = \frac{dx}{\pi\sqrt{1-x^2}};$$
$$F(\mu_0) = \log 2.$$

Now we solve the equilibrium measure under one additional constraint $\int x d\mu \geq r$:

$$\inf_{\mu: \int x d\mu \geq r} F(\mu).$$

Since

$$\int_{-1}^1 d\mu = \frac{T_1(x)}{\pi\sqrt{1-x^2}} \sum_{i=0}^\infty a_i T_i(x) dx = \frac{a_1}{2} = r,$$

this suggests that the equilibrium measure $\mu_r$ is

$$\mu_r(x) = \frac{1 + 2rx}{\pi\sqrt{1-x^2}} \quad (r \leq 1/2),$$

and

$$\inf_{\mu: \int x d\mu \geq r} F(\mu) = F(\mu_r) = \log 2 + 2r^2.$$

Now we are close to completion: our rate function $\tilde{I}(\mu)$ is computed as

$$\tilde{I}(\mu) = F(\mu) - c = F(\mu) - F(\mu_0) = F(\mu) - \log 2.$$

We conclude that the rate is

$$\exp\left(-m^2 \cdot \inf_{\mu: \int x d\mu \geq r} \tilde{I}(\mu)\right)$$

$$= \exp\left(-m^2 \cdot (\inf_{\mu: \int x d\mu \geq r} \tilde{F}(\mu) - F(\mu_0))\right)$$

$$= \exp\left(-m^2 \cdot 2r^2\right),$$

which shows a good rate function for $r$: $I(r) = 2r^2$.

The function $\mu_r$ induces a measure if and only if $r \leq 1/2$. When $r > 1/2$, the function $\mu_r$ no longer induces a measure, as it violates positivity near $x = -1$. This suggests that our theoretical bound, $\log 2 + (2r)^2$, cannot be attained here (To satisfy the positivity of $\mu$, the coefficients $a_2, a_3, \cdots$ cannot all be zero simultaneously).

Overall, we have

$$\inf_{\mu: \int x d\mu \geq r} \tilde{I}(\mu) \geq (2r)^2$$

and

$$\begin{cases} I(r) = 2r^2, & \text{for } r \leq \dfrac{1}{2}; \\ I(r) > 2r^2, & \text{for } r > \dfrac{1}{2}. \end{cases}$$

$\square$

## D.2 ADDITIONAL NOTES ON THE $O(2m) \setminus SO(2m)$ CASE

When $A \in O(2m) \setminus SO(2m)$, $\det(A) = -1$. There are two eigenvalues fixed at $+1$ and $-1$, respectively.

Our formula becomes:

$$p(t_1, \cdots, t_{m-1}) dt_1 \cdots dt_{m-1} = C' \prod_{1 \leq k < j \leq m-1} (t_k - t_j)^2 \cdot \prod_{1 \leq i \leq m-1} (1 - t_i^2)^{+1/2} dt_1 \cdots dt_{m-1}.$$

Taking the logarithm, we have

$$-\log p(t_1, \cdots, t_{m-1}) = \sum_{1 \le k < j \le m-1} (-2 \log |t_k - t_j|) - \sum_{1 \le i \le m-1} \left( \frac{\log(1 - t_i^2)}{2} \right) + C_0.$$

The only differences from above are that $m$ is replaced by $m - 1$ and the external field term is negated. Both changes are negligible in the thermodynamic limit $m \to \infty$. Thus, we obtain the same conclusion for the rate function.

### D.3 THE EXACT FORM OF RATE FUNCTION $I(r)$

We conjecture that the exact form of $I(r)$ is:

$$\begin{cases} I(r) = 2r^2, & \text{for } r \in \left( 0, \frac{1}{2} \right]; \\ I(r) = \frac{1}{2} - \log 2 - \log(1 - r), & \text{for } r \in \left[ \frac{1}{2}, 1 \right). \end{cases}$$

Like the Gross-Witten Ensemble (Gross & Witten, 1980), there is a third-order phase transition near $r = 1/2$.

## E ADDITIONAL EXPERIMENTS

### E.1 COMPARISON WITH REEF

We benchmark our method against REEF (Representation Encoding Fingerprint) (Zhang et al., 2025) on two model pairs:

- **Llama-3.1-8B vs. Llama-3.2-1B** (Meta-AI, 2024): A pruned version where we expect a structured layer correspondence;
- **Llama-3.1-8B vs. Qwen3-8B-Base** (Yang et al., 2025): Two independently developed models with no known lineage.

Our results demonstrate that MDIR is capable of interpretably reconstructing layer mappings between related model weights. In Figure 8(a), MDIR produces a diagonal-like pattern, indicating that each layer of Llama-3.2-1B aligns with a specific layer in Llama-3.1-8B (the 16 layers of Llama-3.2-1B are derived from layer 0, 1, 2, 3, 4, 5, 8, 11, 14, 17, 20, 23, 26, 29, 30 and 31 of Llama-3.1-8B, respectively). This reveals not only homology but also the exact strategy used during model pruning. This offers MDIR a very fine-grained level of interpretability which previous methods cannot provide.

In contrast, REEF (Figure 8(c)) yields uniformly high CKA similarity measures across many pairs of layers and failing to identify the exact correspondence. While it captures global similarity, it lacks the granularity to reveal which layers are actually aligned, thus less effective for model homology detection.

When applied to unrelated models (Figure 8(b,d)), MDIR correctly outputs a noise pattern, consistent with the absence of structural homology. REEF, however, continues to report high similarities ($> 0.9$) between certain layer pairs.

### E.2 ANALYSIS OF A SPECIAL CASE

We present a case study of Qwen2.5-14B (Qwen et al., 2025) and Pangu-pro-MOE (Ascend Tribe, 2025; Ascend Team, 2025) in Figure 9.

## F GENERATIVE AI USAGE

We used generative AI tools, including large language models such as ChatGPT and Qwen, to assist with language editing and generate the code for figures visualizing experimental results. Specifically,

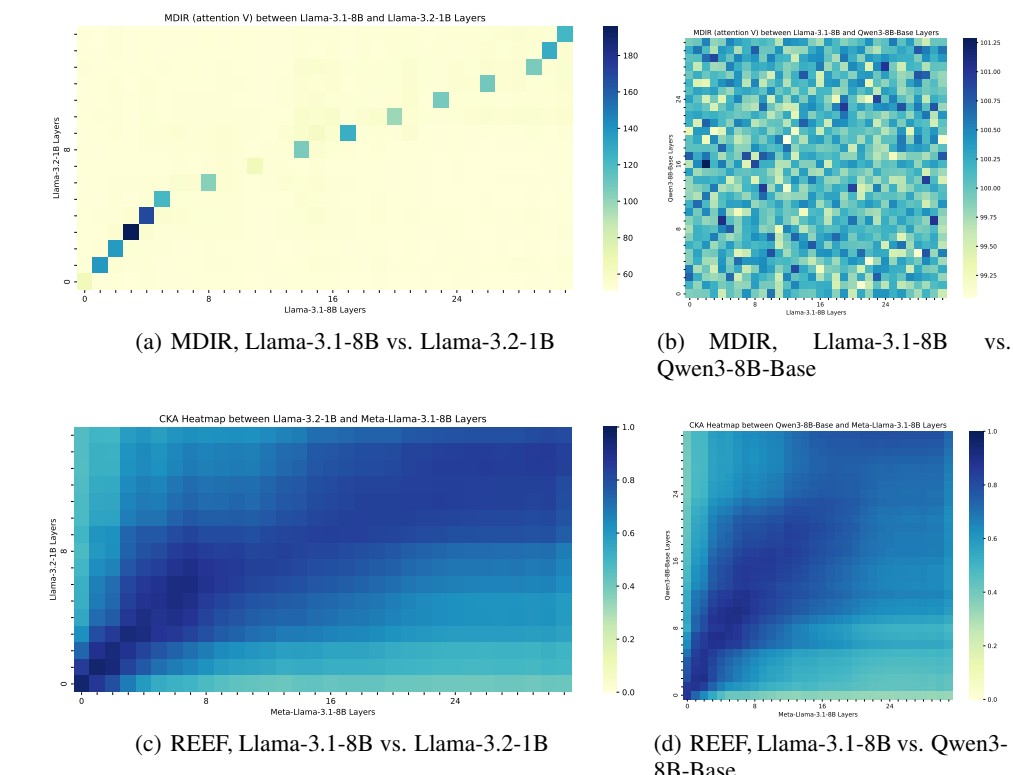

(a) MDIR, Llama-3.1-8B vs. Llama-3.2-1B

(b) MDIR, Llama-3.1-8B vs. Qwen3-8B-Base

(c) REEF, Llama-3.1-8B vs. Llama-3.2-1B

(d) REEF, Llama-3.1-8B vs. Qwen3-8B-Base

Figure 8: MDIR vs. REEF on some test cases.

ChatGPT and Qwen-VL were used to improve the clarity and fluency of the manuscript's writing (including translation of some sections from the original manuscript written in a different language), and to write code to produce plots based on the original data. All scientific content, including the methodology, experimental design and mathematical proof, was written originally and validated by the human authors. The use of generative AI was limited to drafting assistance and visualization support, and no AI system contributed to the core intellectual ideas or conclusions of this work. The authors take full responsibility for the accuracy and integrity of all content presented in this paper.

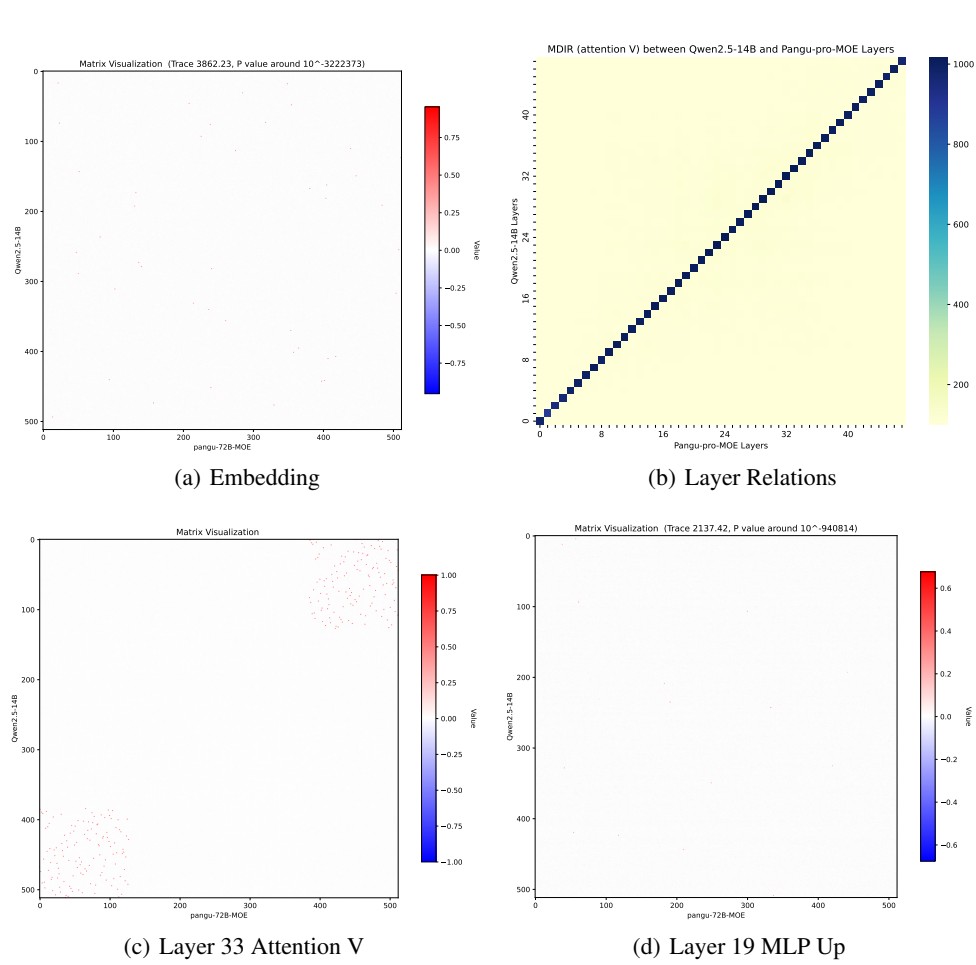

(a) Embedding

(b) Layer Relations

(c) Layer 33 Attention V

(d) Layer 19 MLP Up

Figure 9: A case study of Qwen2.5-14B and Pangu-pro-MOE. For large matrices, we plot the first $512 \times 512$ submatrix.

