# OpenReview forum: "Matrix-Driven Detection and Reconstruction of LLM Weight Homology"
_ICLR.cc/2026/Conference — Submitted to ICLR 2026_

### Official Review · Reviewer_zuU1 · 2025-10-15

**Soundness:** 3
**Presentation:** 2
**Contribution:** 3
**Rating:** 4
**Confidence:** 4

**Summary:**

The paper proposes MDIR, a method for detecting weight homology between LLMs using matrix analysis and Large Deviation Theory. The approach analyzes weight matrices directly to identify if models share common ancestry through transformations like pruning, fine-tuning, or permutations, providing rigorous p-value estimates without requiring model inference.

**Strengths:**

- The mathematical framework connecting invariant transformation groups, polar decomposition, and Large Deviation Theory to LLM weight detection appears novel, with detailed theoretical derivations provided in the appendices.

- The method demonstrates interpretability by reconstructing specific transformations (e.g., Figure 8a shows layer correspondence in Llama-3.2-1B pruning), which could provide insights beyond binary homology detection.

**Weaknesses:**

- The theoretical foundation relies on assumptions that may not hold in practice. The method assumes training dynamics preserve G-coordinates under "idealized conditions (infinite numerical precision and G-invariant optimizers)" (line 131), but modern training uses Adam/AdamW (not G-invariant) and low-precision formats (fp16/bf16). While the authors note "up to 1% discrepancy" between fp32 and fp64 (line 480), there is no systematic analysis of how Adam optimization or quantization affects the invariance assumptions. The claim that "α remains constant at its initialization value" (line 134) would benefit from empirical validation across different optimizers, learning rates, and training durations.

- The evaluation of false positive and false negative rates appears limited. The study tests only 25 models primarily from 4 families (Llama, Qwen, DeepSeek, RWKV) without systematic analysis of negative pairs from diverse architectures. The threshold p < 2×10^{-23} is stated but not justified through ROC analysis or comparison with alternative thresholds. The paper would benefit from: (1) testing on models deliberately trained to be similar (e.g., same architecture and data, different seeds) to establish false positive rates, (2) varying the similarity threshold to construct precision-recall curves, and (3) explaining whether the extreme p-values (10^{-104}) reported are necessary or if detection could work with less extreme thresholds.

- The robustness to adversarial evasion is not empirically tested. Section 5 acknowledges "potential ways of evading detection" as future work, but the paper only evaluates models not designed to avoid detection. Critical untested scenarios include: (1) deliberately retraining embedding layers with higher learning rates to break the orthogonal relationship, (2) applying non-orthogonal transformations before fine-tuning, (3) adding strategic noise to weights. The claim that the method is "exceedingly difficult for adversaries to bypass" (line 71) would be strengthened by adversarial experiments. Additionally, the changed tokenizer case (Section 3.2) assumes shared tokens retain aligned embeddings, which may not hold if embeddings are retrained.

- The experimental comparison with existing methods appears incomplete. Section E.1 compares with REEF on only 2 model pairs through visual inspection, without quantitative metrics (accuracy, F1, precision-recall). The ablation study (Section 4.3) tests only same-seed initialization on two small datasets, but does not systematically vary: (1) amount of training data (would 100B tokens break detection?), (2) different learning rates, (3) continued pretraining duration, (4) mixing weights from multiple models. The computational cost is stated as "single consumer PC" but no wall-clock time or memory comparisons with baselines are provided. For the GQA transformation group construction (Section 3.1), the paper states this is a "sufficient subset" but does not prove completeness—unexplored transformations could enable evasion.

I will reconsider my score in the rebuttal.

**Questions:**

see weaknesses

---

> ### Author Response · Authors · 2025-11-20
>
> The motivation section in the paper is not intended to serve as the theoretical foundation of our method. It is written to facilitate readers' understanding and provide some intuition on how we come up with the proposed formulation. The theoretical soundness of our method comes from the mathematical formula itself:
> $$
> \max_{P}\mathrm{Tr}(P\cdot\mathrm{Ortho}(E'^T E)^T).
> $$
> This formula is grounded in the geometry of the Orthogonal Procrustes Problem and the Large Derivation Theory behind random orthogonal matrices.
>
> We would also like to clarify that α comes from an implicit local coordinate system, which does not have a canonical representation. It is not possible to reconstruct the true value of α due to the inherent ambiguity in choosing a local coordinate basis. This discussion is included solely to provide some intuition, not for the actual computation.
>
> ---
>
> To analyze false positive and false negative rates, we require a large number of ground-truth model pairs. Obtaining such data is challenging due to the lack of available information (e.g., our method shows homology between Qwen2-7B and Qwen2.5-7B. This is undocumented in Qwen's technical report, yet we cannot rule out this possibility). Additionally, some companies may fabricate technical reports, complicating the collection of ground-truth relationships.
>
> Also, we strive to design our method to be calibration-free, and the $p$-value threshold is left as the last hyperparameter. In practice, we observe that the $p$-values produced by our method tend to fall into two distinct cases: either close to 1 (not homologous) or astronomically small (strong correspondence). As a result, the choice of threshold within a wide range (e.g., $10^{-3}$ to $10^{-100}$) will have little impact on the final results.
>
> ---
>
> (1) The "orthogonal relationship" is computed from the orthogonal factor from polar decomposition. This is mathematically defined and cannot be broken.
>
> (2) It is almost impossible to enumerate all transformations that preserve model functionality (this could include non-orthogonal, non-linear or even discontinuous transformations). An adversary may apply some unexplored transformations, but probably at the cost of model functionality.
>
> (3) See supplementary experiment (Adversarial Noise Incapacitates Models Before Evading MDIR) above.
>
> Additionally, see supplementary experiment (Continual Pretraining with a Modified Tokenizer).
>
> ---
>
> > Section E.1 compares with REEF ...
>
> REEF provides an $a \times b$ matrix when comparing a model with $a$ layers to another with $b$ layers. Extracting one similarity score from this matrix is non-trivial, especially when $a \ne b$. Moreover, it does not provide binary outputs.
>
> > without quantitative metrics
>
> Obtaining the ground truth is challenging.
>
> > (1)-(4)
>
> Exhaustive testing to address each point would require experiments at exponentially larger scales.
>
> > The computational cost
>
> We invite you to run the attached code. See also our response to Reviewer EF5y, point 3.
>
> > does not prove completeness
>
> As noted earlier, it is almost impossible to enumerate all transformations that preserve model functionality.

---

### Official Review · Reviewer_h2j3 · 2025-10-24

**Soundness:** 2
**Presentation:** 2
**Contribution:** 3
**Rating:** 4
**Confidence:** 2

**Summary:**

The paper studies model weight 'homology'. In particular, they propose a method to infer whether a model parameterized by theta_a is derived from another model theta_b. Authors consider a broad range of what constitutes 'derivation', including continual pretraining, finetuning, pruning etc (5 in total, and a combination of all).

They propose a mathematically grounded method called MDIR to compare the similarity between the matrices of parameters, leveraging SVD and polar decomposition. Instead of setting a threshold for this similarity metric, or learning it using some ground truth, they further use Large Deviation Theory (LDT) and random matrix theory to estimate p-values (where the null hypothesis is that both models are not homologous).

Authors then consider 25 open-source models and use MDIR to estimate whether each combination of these models can be considered homologous with respect to a certain p-value. They find their method to give strong signal for models with a known homology relationship, while giving no signal for independently developed models.

They further add an ablation experiment studying whether their method picks up on weight initialisation versus training data similarity, confirming that it is the former.

As a **disclaimer,** I want to clarify that, as a reviewer, my mathematical background likely does not suffice to thoroughly critique or fully appreciate the proposed method in Section 3. I therefore focus most of my review on the motivation, experimental setup, results and conclusions.

**Strengths:**

- The problem of identifying model homology is interesting, and the paper proposes a novel way, grounded in mathematics, to address this.
- The fact that the method allows for the computation of a p-value a priori (no ground truth is needed) is valuable, and remains overlooked by other methods.
- The ablation experiment in section 4.3 is very nice, and cleanly distinguishes between initialization and training data similarity.

**Weaknesses:**

- Authors do not consider any baselines, including some representation-based methods from prior work. I understand from the introduction that these methods "generally lack the ability to reconstruct the weight correspondence mapping", but when it comes to vizualizing the similarity metric as in Figure 3a, these methods could be presented as a valuable baseline. Moreover, to justify the mathematical complexity of the method, some more naive baselines (e.g. norm of the difference between the weights) would further ground the significance of the results.
- While I find Section 2.2 easy to follow, Section 3 quickly becomes hard to understand. The paper could be appreciated by a wider audience if the most relevant pieces of the method remain in the main body of the paper, while the other pieces could be put in the appendix. On that note, it might also be useful to summarize the core of the method either at the end or at the start of section 3.
- Sections 4.1 and 4.2 would benefit from more elaborate description of the results and what they mean.

**Questions:**

- What do authors mean by "Independently developed models" on line 366?
- In figure 3B, could you explain why there seems to be homologous relationship between Qwen-2.5-14B and Pang-Pro-MOE?
- An interesting set of additional experiments would be to use the pretrained models from section 4.3 and apply multiple transformations to their weights (e.g. continued pretraining, SFT+DPO, pruning) and evaluate how the similarity metric and p-value compare across transformations, and for e.g. dataset size or hyperparameters in the transformations. Such results would further illustrate the robustness of the method.

---

> ### Author Response · Authors · 2025-11-20
>
> ## Weaknesses
>
> We appreciate your suggestion regarding additional baselines. However, we would like to clarify that we don't include baselines not because we are unwilling to, but because our method differs from prior representation-based approaches in both its goals and mathematical meanings.
>
> Our approach operates at the weight level rather than the representation level. While representation-based methods such as REEF focus on comparing intermediate representations using metrics like Central Kernel Alignment (CKA), our method directly analyzes the statistical significance of weight correspondence through Large Deviation Theory (LDT) derived from a trace formula: $ T(E, E') = \max_{P} \mathrm{Tr}\left(P \cdot \mathrm{Ortho}\left(E'^{\text{T}} E\right)^{\text{T}}\right)$.
> The trace is converted into a $p$-value using LDT, which gives the statistical significance of weight alignment. Prior methods, especially those based on similarity metrics, do not provide a probabilistic interpretation. Although it is possible to normalize our trace formula into a similarity metric (like by dividing by the maximum possible value, i.e., embedding dimension), this may obscure mathematical or physical meaning. Our primary goal is not to produce a similarity score, but to evaluate the statistical significance of weight mappings. Therefore, while Figure 3a presents trace values for illustration, Figure 3b showing $p$-values is more essential.
>
> There are also challenges in applying baseline methods, for example:
>
> 1. The norm of the difference between weights, or cosine similarity, are only limited to cases where the weight matrices have identical dimensions, and these naive metrics cannot handle models with different architectures or layer sizes.
>
> 2. REEF computes pairwise similarity between layers using CKA, providing an $a \times b$ matrix when comparing a model with $a$ layers to another with $b$ layers. Extracting one similarity score from this matrix is non-trivial, especially when $a \neq b$. Furthermore, representation-based methods like REEF rely on intermediate representations, requiring full inference on the entire model, which is beyond our computational resources for very large models (1T parameters).
>
> Another challenge is that prior baseline methods lack a clear decision rule for producing binary outputs (YES/NO) to indicate whether two models are similar or not. Similarity metrics (like CKA or cosine similarity) provide similarity scores without a systematic way to threshold these values. Our method, on the other hand, utilizes the $p$-value derived from the trace formula to provide a statistical decision rule. This allows us to make binary decisions with error rate thresholds. Without a comparable decision mechanism, it is unclear how to benchmark our method against these baselines.
>
> ---
>
> ## Questions
>
> 1. By "Independently developed models", we mean neither model is trained from another, nor do any part of their weights share a common ancestor. It means that they are initialized and trained separately without shared initialization, pre-trained weights, or intermediate checkpoints. Therefore, if a model is initialized with its unique seed and trained independently from scratch, it is considered "independently developed" with respect to all other models.
>
>
> 2. Our method reveals homologous relationship between Qwen-2.5-14B and Pangu-Pro-MOE in Figure 3B is indeed intriguing but difficult to explain definitively without official confirmation from the Pangu team. Our experiments yield a large trace of alignment and astronomically low $p$-value, despite the two models using different tokenizers. If this is true, it might suggest that Pangu-Pro-MOE is upscaled from Qwen-2.5-14B with a modified tokenizer (see also one of the supplementary experiments), while the non-identity permutation pattern observed during channel alignment hints at the possibility of intentional obfuscation. Considering the societal implications (not to raise too much public attention to comply with anonymity policy), we apologize that our initial paper may not have explained the details properly.
>
>
> 3. Such experiments would require significant computational budgets. Nevertheless, our current results already demonstrate the robustness of our method in detecting homology even after extensive training. For instance, we have shown that our approach can identify weight correspondence and statistical significance (via p-values) for models trained with trillions of additional tokens (e.g., Qwen2-57B-A14B is upscaled with 4.5T tokens). For some additional Base/SFT pairs, like Qwen2.5-0.5B and Qwen2.5-0.5B-Instruct, Qwen3-8B-Base and Qwen3-8B, however, there is such a subtle difference that will hardly affect our detection.

---

> > ### Comment · Reviewer_h2j3 · 2025-11-24
> > **Response to author rebuttal**
> >
> > Many thanks to the authors for their rebuttal.
> >
> > Some of my concerns (e.g. regarding baselines) have been addressed, while others (such as providing more explanation in Section 4 and conducting more extensive experiments) remain unresolved. I am therefore keeping my original score.

---

### Official Review · Reviewer_6fc4 · 2025-10-29

**Soundness:** 3
**Presentation:** 3
**Contribution:** 2
**Rating:** 6
**Confidence:** 3

**Summary:**

The paper introduces Kernelized Dynamics Pruning (KDP), a new method for pruning layers in large language models by viewing their forward computation as a discrete-time dynamical system. It observes that consecutive layers often produce highly similar representations, indicating redundancy. To exploit this, KDP projects representations into a kernel space where nonlinear transformations become approximately linear, allowing simpler modeling. A linear operator and an inverse mapping network then replace entire Transformer blocks. The authors present a theoretical error bound showing that multi-layer dynamics can be linearly approximated in this kernel space and prove that it provides superior fitting capacity compared to the original representation space. Extensive experiments on fifteen benchmarks demonstrate that KDP effectively prunes models while maintaining performance, without requiring fine-tuning. Overall, KDP provides a geometric and theoretically principled framework for simplifying internal model dynamics and reducing redundancy in large language models.

**Strengths:**

- Originality: Previous methods rely either on vendors implanting specific proprietary key-value pairs, or similarity measures of representations/weights. This work operates along the line of the second method, but also identifies the transformation involved (based on known symmetries of the weight matrices), which allow for a more fundamental detection of similarity. Moreover, they improve the statistical soundness of evaluating detection through a better evaluation of p-value based on large deviation theory arguments.
- Quality: The theoretical part seems solid. The experiments are extensive, showing the method for a large number of examples. The algorithm is computationally efficient and runs on a laptop, allowing broad use. The appendix also includes comparison with previous methods (REES).
- Clarity: The problem to solve is well posed and the writing is understandable and linear.
- Significance: The method seems to improve significantly the state-of-the-art of identifying unattributed reuse of LLMs.

**Weaknesses:**

As a general comment, it is not completely clear to me the extent to which weight relationships are included in the algorithm. The authors list a few of them, saying the list is not exhaustive. This is fine, as it is probably hard to write down an exhaustive list, but then I would expect that at least mathematically we have a clear statement of which groups of transformations satisfy the hypotheses of the method. The authors say that “we don’t need to characterize all totally invariant transformations; We need a subset with sufficiently large dimension,  which is enough for subsequent analysis with high confidence”. This is in principle explained in section 3.1, but I don’t see this specific point clearly adressed. Since this is a key point of the papers, the authors might consider refining this part to improve the manuscript.

**Questions:**

- Along the line of the comments above, why are transformations from theta_A to theta_B being considered linear (cfr. line 176) ?
- Just below, the authors show that Uq, Uk and Uv must be orthogonal matrices, but they don’t say anything about Uo. Should we consider this to be a generic matrix?
- The subset of W transformations is declared to be sufficient on lines 196-200. Is this proven?

---

> ### Author Response · Authors · 2025-11-19
> **Clarification of Section 3.1**
>
> Thank you for your constructive feedback. We agree that a clearer mathematical statement would further clarify the groups of transformations.
>
> In Section 3.1, we describe the invariant transformations of the model parameters under certain symmetries. Specifically, the transformation group $G$ is generated by the Cartesian product of the following components:
>
> $$
> U, \ (P_{1,\ell})_{\ell=1..L}, \ (P_{2,\ell})_{\ell=1..L}, \ (S_{\ell})_{\ell=1..L}, \ (H_{\ell,v})_{\ell=1..L, v=1..\mathrm{NumKVHeads}},
> $$
>
> where each component satisfies the constraints outlined in the paper (line 230-239).
> - $U$ represents the outer transformation, which is shared across all layers and belongs to the orthogonal group $\mathrm{O}(\mathrm{EmbDim})$;
> - $(P_{1,\ell})$ and $(P_{2,\ell})$ are permutation matrices corresponding to attention heads and queries respectively;
> - $(S_{\ell})$ are diagonal matrices with entries $\pm 1$ to ensure compatibility with Rotary Positional Embeddings;
> - $(H_{\ell,v})$ are general invertible matrices applied to individual attention heads.
>
> The group $G$ is defined as the Cartesian product of these components:
>
> $$G = ( U )  \times \prod_{\ell=1}^L \left( (P_{1,\ell}) \times (P_{2,\ell}) \times (S_\ell) \times \prod_{v=1}^{\mathrm{NumKVHeads}} \(H_{\ell,v}) \right)$$
>
> This captures some of the possible combinations of transformations that preserve the model's functionality.
>
> It is almost impossible to enumerate all transformations that preserve model functionality (this could include non-orthogonal, non-linear or even discontinuous transformations). However, the subset we propose is sufficiently large for our subsequent analysis, because it spans a high-dimensional space of transformations, facilitating accurate estimation via Large Deviation Theory.
>
> We are planning to revise Section 3.1 to include a more explicit definition of the transformation group $G$ using Cartesian product notation. Please let us know if further elaboration is needed.
>
> ## Questions
>
> > why are transformations from theta_A to theta_B being considered linear
>
> It is almost impossible to enumerate all transformations that preserve model functionality — this could include non-orthogonal, non-linear or even discontinuous transformations. We consider linear transformations for simplicity, yet the group structure is already relatively complex.
> It is possible if an adversary uses additional transformations, for example:
> - Linear interpolation, if two neurons (by coincidence) have nearly the same weights;
> - Some unexplored discontinuous transformations, but probably at the cost of model functionality.
>
> > Just below, the authors show that Uq, Uk and Uv must be orthogonal matrices, but they don’t say anything about Uo
>
> We apologize for the lack of clarity regarding $U_O$. In fact, $U_O$ is not a generic matrix but is directly related to $U_Q, U_K,$ and $U_V$. Specifically:
> $$
> U_Q = U_K = U_V = U_O^T,
> $$
> where all these matrices are orthogonal. The orthogonality of $U_Q, U_K, U_V$ is required because they operate on normalized vectors (via RMSNorm). To maintain consistency across layers, $U_O$ must also adhere to the same constraint, specifically as the transpose of $U_Q$. Please let us know if further clarification is needed.
>
>
> > The subset of W transformations is declared to be sufficient on lines 196-200. Is this proven?
>
> To clarify, the term "sufficient" refers to the structural adequacy, not the mathematical sufficiency condition.
>
> The transformations cover a wide range of commonly used obfuscation methods, such as permutations, rotations, and scaling. The subset provides a foundation for applying Large Deviation Theory (LDT) and other analytical tools in our framework with high confidence, even if it does not include all possible transformations.
>
> ---
>
> In general, we agree with your suggestions and will revise Section 3.1 to address these concerns.

---

> > ### Comment · Reviewer_6fc4 · 2025-11-20
> > **Official Reply to Rebuttal**
> >
> > I thank the authors for replying carefully to my concerns point by point. I believe they addressed all of them, and I do not have further comments.

---

### Official Review · Reviewer_EF5y · 2025-11-01

**Soundness:** 3
**Presentation:** 2
**Contribution:** 3
**Rating:** 4
**Confidence:** 3

**Summary:**

To detect LLM weight homology, this paper proposes Matrix-Driven Instant Review (MDIR), leveraging matrix analysis and Large Deviation Theory to identify weight reuse, reconstruct transformation relationships between weights, and provide rigorous p-value estimates.

**Strengths:**

- The proposed method is novel and contributes to the protection of open-source model IP.
- Experiments demonstrate the robustness and practicality of the proposed method.

**Weaknesses:**

- The paper's core motivation relies on an "idealized" assumption (Lines 106-107). Is there related experimental evidence or research to support this?
- MDIR depends on first reliably computing $U$ from the embedding layers $E$ and $E'$. If an adversary intentionally injects large-scale, non-orthogonal noise into or retrains only the embedding layers, it could lead to an incorrect $U$ calculation, causing all subsequent layer detections to fail. There is a lack of discussion on the lower bounds of MDIR's robustness.
- There are potential overclaims, such as an undefined "consumer PC" and no discussion on computational resource consumption.
- The method is not applicable to closed-source models. In real-world scenarios, malicious users often provide API services, which limits its application.
- There are clarity issues with the paper's presentation, such as excessively small text in figures (e.g., Figures 4, 5, 6, and more) and an illogical presentation of text content in Section 4.2.

**Questions:**

- If multiple transformations (e.g., permutation and quantization) exist in the model simultaneously, can MDIR reconstruct them?
- Why is there no specific pattern in Figure 5(c)? The pattern in Figure 4(c) is also unclear. Is this related to the properties of MLPs?
- Is this method applicable to model merging scenarios?

---

> ### Author Response · Authors · 2025-11-19
>
> 1 The motivation section in the paper is not intended to serve as the theoretical foundation of our method. It is written to facilitate readers' understanding and provide some intuition on how we come up with the proposed formulation. The theoretical soundness of our method comes from the mathematical formula itself:
> $$
> \max_{P}\mathrm{Tr}(P\cdot\mathrm{Ortho}(E'^T E)^T).
> $$
> This formula is grounded in the geometry of the Orthogonal Procrustes Problem and the Large Derivation Theory behind random orthogonal matrices.
>
> ---
>
> 2 We appreciate your concern regarding the robustness. While adversarial noise may lead to incorrect computation of $U$, please note that such attacks may also incapacitate the language model itself.
>
> In a supplementary experiment, we show that an adversary cannot preserve model utility while evading MDIR by injecting random noise. When injecting noise at 3x the RMS of the original embeddings, the performance of the language model collapses (shown by astronomically large perplexity and near-random MMLU accuracy). Yet, our MDIR method still reconstructs the correspondence across channels and produces extremely significant p-values.
>
> ---
>
> 3 We appreciate your concern about computational resource consumption. However, we did not intend to overclaim the computational requirements. We welcome you to verify our claims by running the supplementary code provided with the paper on your own machine.
>
> For reference, here is the configuration of our testing environment:
> - CPU: AMD Ryzen 9 7950X
> - GPU: NVIDIA RTX 4090 (24GB VRAM)
> - Memory: 128GB DDR5 RAM
> - Disk: 4TB SSD
>
> Our code may also be run on machines with lower specifications.
>
> The computational complexity of our method is $O(n^3)$, where $n$ is the embedding dimension of the model (not the total number of parameters). Typically, $n$ does not exceed 10,000. The main steps in computation are as follows:
> 1. Matrix multiplication of the vocabulary embeddings: $O(n^3)$
> 2. Polar decomposition to extract the orthogonal matrix: $O(n^3)$
> 3. Solving the linear sum assignment (maximum matching) problem: $O(n^3)$
>
> Given this complexity, our method can be efficiently run on consumer-grade PCs in reasonable time (likely less than a minute for a single pair).
>
> ---
>
> 4 We acknowledge your concern that our method is not directly applicable to closed-source models, where only API access is granted. However, this limitation is not unique to our method, but a common challenge for model homology/fingerprinting methods that inspect model parameters or internal representations. We believe our method still has value in circumstances where model weights are accessible (either open-source models or internal audits of proprietary models).
>
> ---
>
> 5 We agree that the text in some figures is excessively small, and the logical flow in Section 4.2 could be improved. We will revise the structure and content of Section 4.2 to improve its logical flow and clarity.
> Meanwhile, we would greatly appreciate it if you could provide more specific suggestions on how to improve the presentation.
>
> ---
>
> ## Questions
>
> 1 Yes, MDIR can reconstruct multiple transformations simultaneously. Here are the reasons:
> - Permutation-invariance: Our method is proven (see supplementary) to be invariant to channel permutations.
> - Quantization noise is typically small relative to the root-mean-square (RMS) of the original embeddings. Also, vocabulary embeddings are rarely quantized.
> - Experimental Evidence: For example, MDIR successfully detects relationships between Llama-3.1-8B, Llama-3.2-1B, and Llama-3.2-3B, which involve pruning and further pretraining on up to 9 trillion tokens.
> - Transitivity: Since our method detects model homology (shared origins), it satisfies transitivity: if Model A is homologous to Model B, and Model B is homologous to Model C, then Model A is also homologous to Model C. This ensures consistency across transformations.
>
> ---
>
> 2 We have uploaded a high-resolution version of Figure 5(c) in the supplementary materials. The red pixels, which constitute approximately 1/8192 of the total (the MLP has an intermediate dimension of 8192), are hardly distinguishable due to their sparsity. The pattern indicates irregular, non-sequential correspondence between the MLP channels. This suggests that during the upscaling process, some channels in the MLP layers may have been rearranged or reselected.
>
> ---
>
> 3 The applicability of our method to model merging scenarios depends on the specific definition of "model merging." Generally, merging language models requires some prerequisites, such as having identical architectures and tokenizers. Moreover, merged language models are often fine-tuned from a common base model through SFT/RLHF, which implies that they are already homologous. If you are referring to merging models from entirely different origins, to the best of our knowledge, there are no successful practices in the industry for such scenarios.

---

### Author Response · Authors · 2025-11-14
**Added Supplementary Material**

We apologize for the oversight in not including the supplementary code in the submission.
Now, we have uploaded supplementary material, including the source code and the full-resolution Figure 5(c), in hope to address some of the concerns.
All experiments in the paper are reproducible via the provided code.
We are actively conducting additional experiments to address the reviewers’ concerns. Meanwhile, we greatly appreciate any specific suggestions on further experiments that would be most useful.
We aim to address each concern raised in the reviews in the next few days.

---

### Author Response · Authors · 2025-11-14
**Robustness: Adversarial Noise Incapacitates Models Before Evading MDIR**

We are writing to address Reviewer EF5y’s concern regarding the robustness of MDIR to adversarial perturbations in the embedding layers, as well as Reviewer h2j3's concern on missing baselines. We appreciate the opportunity to address this point with evidence from some experiments.

We directly tested the robustness of MDIR by injecting increasing levels of Gaussian noise into the embedding (and unembedding, when they are not tied) matrices of two models: Qwen3-0.6B-Base and Llama-3.2-3B. The noise was scaled as  `noise_level * RMS(original_matrix)` to ensure fair comparison across scales. We evaluated:
- Our method’s trace and normalized trace (the trace divided by the model dimension)
- Two baseline similarity measures: naive cosine similarity and REEF’s Central Kernel Alignment (CKA)
- The functionality of the model via LAMBADA perplexity and MMLU accuracy

The results are summarized below:

### Qwen3-0.6B-Base
| Noise Level | Trace (MDIR) | Normalized Trace | Cosine Sim. | CKA (REEF) | LAMBADA PPL | MMLU Acc |
|-------------|--------------|-------------|-------------|------------|--------------|----------|
| 0x (original)          | 1024         | 1       | 1       | 1      | 9.6          | 50.4     |
| 0.1x        | 1023.98      | 0.99998     | 0.995       | 0.997      | 11.0         | 49.7     |
| 0.3x        | 1023.82      | 0.9998      | 0.958       | 0.973      | 24.7         | 40.0     |
| 1x          | 1021.96      | 0.998       | 0.707       | 0.582      | 31,412       | 27.1     |
| 3x          | 1005.6       | 0.982       | 0.316       | 0.291      | 4e16       | 25.7     |

### Llama-3.2-3B
| Noise Level | Trace (MDIR) | Normalized Trace | Cosine Sim. | CKA (REEF) | LAMBADA PPL | MMLU Acc |
|-------------|--------------|-------------|-------------|------------|--------------|----------|
| 0x (original)         | 3072         | 1            | 1               | 1           | 3.95          | 54.1     |
| 0.1x        | 3071.77      | 0.99992     | 0.995       | 0.9986     | 3.98         | 53.7     |
| 0.3x        | 3069.88      | 0.9993      | 0.958       | 0.985      | 4.36         | 51.3     |
| 1x          | 3048.44      | 0.992       | 0.707       | 0.721      | 37.9         | 26.0     |
| 3x          | 2859.64      | 0.931       | 0.316       | 0.455      | 5e12       | 24.9     |

Results show that: firstly, MDIR is more robust than baselines. Even at 3x noise (subsequent fine-tuning and quantization are unlikely to produce changes at this scale), MDIR retains more than 93% of maximum possible trace value, while naive cosine similarity and CKA drops significantly to around 0.3~0.5. This demonstrates that MDIR preserves structural signal far better than standard similarity metrics under strong perturbation.

More crucially, noise destroys model functionality before detection fails. At 1x noise, both models suffer catastrophic performance collapse (MMLU drops to near-random). Yet, MDIR still reports a normalized trace greater than 0.99, indicating detection remains highly confident.

---

Meanwhile, we are planning to conduct more experiments, such as continual pre-training after modifying an existing model's tokenizer, to verify that MDIR remains operational under these scenarios. Will this experiment address your concern? We look forward to your valuable suggestions on experimental design.

---

### Author Response · Authors · 2025-11-14
**Robustness: Mathematical Proof of Preservation under Scaling and Permutation**

We write this section in hope to address reviewer's concern on the robustness of our method.

---

Following Section 3.2, we let $E$ and $E'$ denote the vocabulary embeddings of model $A$ and $B$, respectively. We demonstrate that the final results remain invariant when an adversary applies both scaling and permutation to the matrix $E'$. Specifically, we consider the transformed embedding $E'' = \alpha E' P'$, where $P' \in \mathrm{Perm}(\mathrm{EmbDim})$ is a permutation matrix, and $\alpha > 0$ is a scaling factor.

## Lemma
Define the trace and the corresponding optimal permutation as follows:

$$
\begin{aligned}
    T(E, E') &= \max_{P \in \mathrm{Perm}(\mathrm{EmbDim})} \mathrm{Tr}\left(P \cdot \mathrm{Ortho}\left(E'^{\text{T}} E\right)^{\text{T}}\right), \quad \\
    \Pi(E, E') &= \arg\max_{P \in \mathrm{Perm}(\mathrm{EmbDim})} \mathrm{Tr}\left(P \cdot \mathrm{Ortho}\left(E'^{\text{T}} E\right)^{\text{T}}\right).
\end{aligned}
$$

Suppose an adversary applies a simultaneous scaling and permutation transformation to $E'$, yielding $E'' = \alpha E' P'$, where $P' \in \mathrm{Perm}(\mathrm{EmbDim})$ is an arbitrary permutation matrix, and $\alpha > 0$ is a positive scaling coefficient. In this case, the following properties hold:

$$
T(E, E'') = T(E, E'), \quad \Pi(E, E'') = P'^{\text{T}} \Pi(E, E').
$$

Thus, such adversarial modifications are futile.

## Proof
We first analyze the trace $T(E, E'')$:

$$
\begin{aligned}
    T(E, E'') &= \max_{P \in \mathrm{Perm}(\mathrm{EmbDim})} \mathrm{Tr}\left(P \cdot \mathrm{Ortho}\left((E''^{\text{T}} E)^{\text{T}}\right)\right) \quad \\
    &= \max_{P \in \mathrm{Perm}(\mathrm{EmbDim})} \mathrm{Tr}\left(P \cdot \mathrm{Ortho}\left(((\alpha E' P')^{\text{T}} E)^{\text{T}}\right)\right) \quad \\
    &= \max_{P \in \mathrm{Perm}(\mathrm{EmbDim})} \mathrm{Tr}\left(P \cdot \mathrm{Ortho}\left(\alpha E^{\text{T}} E' P'\right)\right).
\end{aligned}
$$

Since scaling by $\alpha > 0$ does not affect the orthogonal factor in polar decomposition ($\mathrm{Ortho}$), we can simplify further:

$$
\begin{aligned}
    T(E, E'') &= \max_{P \in \mathrm{Perm}(\mathrm{EmbDim})} \mathrm{Tr}\left(P \cdot \mathrm{Ortho}\left((E^{\text{T}} E') P'\right)\right) \\
    &= \max_{P \in \mathrm{Perm}(\mathrm{EmbDim})} \mathrm{Tr}\left(P \cdot \mathrm{Ortho}(E^{\text{T}} E') P'\right) \\
    &= \max_{P \in \mathrm{Perm}(\mathrm{EmbDim})} \mathrm{Tr}\left(P' P \cdot \mathrm{Ortho}\left(E^{\text{T}} E'\right)\right).
\end{aligned}
$$

Let $R = P' P$. Since $P'$ is a fixed permutation matrix, as $P$ ranges over all permutations in $\mathrm{Perm}(\mathrm{EmbDim})$, so does $R$. Substituting $R$ into the expression, we obtain:

$$
\begin{aligned}
    T(E, E'') &= \max_{R \in \mathrm{Perm}(\mathrm{EmbDim})} \mathrm{Tr}\left(R \cdot \mathrm{Ortho}\left(E^{\text{T}} E'\right)\right) \\
    &= \max_{R \in \mathrm{Perm}(\mathrm{EmbDim})} \mathrm{Tr}\left(R \cdot \mathrm{Ortho}\left((E'^{\text{T}} E)^{\text{T}}\right)\right) \\
    &= T(E, E').
\end{aligned}
$$

Next, we examine the optimal permutation $\Pi(E, E'')$. Recall that the trace achieves its maximum when $R = \Pi(E, E')$. From the substitution $R = P' P$, it follows that:

$$
P' P = \Pi(E, E') \implies P = P'^{-1} \Pi(E, E') = P'^{\text{T}} \Pi(E, E').
$$

Thus, the optimal permutation for $E''$ is given by:

$$
\Pi(E, E'') = P'^{\text{T}} \Pi(E, E').
$$

This completes the proof.

---

We plan to insert this theoretical proof, as well as the previous experimental results, into the Appendix. We welcome reviewers to offer suggestions on the writing style and logical coherence.

---

### Author Response · Authors · 2025-11-18
**Ablation: Continual Pretraining with a Modified Tokenizer**

Following Reviewer zuU1's suggestions, as well as to validate our claim (Lines 282–293) that MDIR detects homology even when the tokenizer and embeddings are replaced, we conduct an ablated experiment in which the base model’s embedding layer is deliberately reinitialized and retrained on a new vocabulary.

We select Qwen2.5-0.5B as the base model (hidden dimension $d = 896$), which originally uses tied embedding and unembedding, with a tokenizer (`Qwen2Tokenizer`) with a vocabulary size 151,665 (padded to 151,936). Then, we discard the original embedding weights, initialize a new embedding matrix of size $50304 \times 896$ with all-zero values (which is trainable because the gradient is non-zero), and continually pretrain the model on the DCLM subsample (see Lines 432–433) tokenized using the GPT-NeoX tokenizer (vocabulary size 50,277, padded to 50,304). Other weight matrices (attention projections, MLPs, layer norms) are inherited from the original Qwen2.5-0.5B (Therefore, we anticipate that MDIR detects model homology). All weights (not just the embeddings) undergo bfloat16 precision training, with no frozen weights. The starting learning rate is $10^{-4}$ with no warm-up, followed by quadratic decay to 0.

We monitor three kay metrics throughout training: the trace $\mathrm{Tr}\left(P \cdot \mathrm{Ortho}\left(E'^{\text{T}} E\right)^{\text{T}}\right)$, the significance of $p$-value, and the number of correctly matched channels (i.e., number of 1's in the permutation matrix $P$ that matches the correct position). The results are summarized in the following table:


| Tokens (B) | Trace | $p$-value | # Correct Channels |
|-------------------|-------|-----------|-------------------|
| 1.64              | 94.5  | NS        | 110               |
| 3.28              | 179.6 | $10^{-4750}$ | 828               |
| 4.82              | 254.6 | $10^{-11813}$ | 896               |
| 6.55              | 296.1 | $10^{-16799}$ | 896               |
| 8.19              | 316.8 | $10^{-19533}$ | 896               |
| 9.83              | 324.3 | $10^{-20584}$ | 896               |
| 11.47             | 325.8 | $10^{-20798}$ | 896               |
| 12.05             | 325.9 | $10^{-20806}$ | 896               |

We observe that, at early training stages (1.64B tokens), the trace is low and the $p$-value is not significant (NS), indicating minimal alignment, since the new embeddings are still undertrained. By 3.28B tokens, the $p$-value plunges to $10^{-4750}$, and 828/896 channels are correctly matched. This is strong evidence that the vocabulary embeddings have almost recovered the channel structure, despite the new tokenizer. At 4.82B tokens or more, All channel relationships are matched, and the $p$-value continues to go down steadily.

Therefore implies that an adversary cannot evade detection of MDIR by continual pretraining with a changed tokenizer.

---

### Meta-Review · Area_Chair_HzKx · 2026-01-07

**Summary:**

This paper proposes to detect weight homology between LLMs using matrix analysis and Large Deviation Theory. The approach analyzes weight matrices directly to identify model derivation relationships without requiring model inference. The reviewers raised several concerns. The reviewers asked about the theoretical foundation may not hold in practice with Adam/AdamW and low-precision training. The authors didn’t provide convincing response on this. The reviewers also raised questions about the experimental evaluation that lacks baseline comparisons, and thorough robustness testing. Based on these weaknesses, the AC suggests a rejection.

**Reviewer Concerns:**

Addressed concerns. Mathematical clarity. The authors provided satisfactory clarification on the transformation group structure and the relationship between transformation matrices. Computational requirements. The authors explained the complexity and provided hardware specifications. Adversarial noise robustness. The experiments showed that noise injection destroys model functionality before evading detection.

Outstanding concerns. Theoretical assumptions. The reliance on idealized conditions that don't hold in practice (Adam optimizer is not G-invariant, training uses fp16/bf16) remains a concern. The authors' response that the motivation section is only for intuition does not fully resolve whether the core method is affected. More adversarial testing. Beyond noise injection, critical scenarios like deliberately retraining embeddings with high learning rates or applying non-orthogonal transformations before fine-tuning were not empirically tested.

**Reviewer Scores:**

Reviewer h2j3 have concerns regarding more extensive experiments, and would remain the score of 4. For Reviewer ef5y, the authors addressed concerns about robustness and computational requirements with supplementary experiments and detailed explanations. However, concerns about  closed-source applicability remain. The score would likely increase to 5.

---

### Decision · Program_Chairs · 2026-01-26

Reject